# Self-organization of modular network architecture by activity-dependent neuronal migration and outgrowth

**Samora Okujeni[1,2]\*, Ulrich Egert[1,2]**

[1]Laboratory for Biomicrotechnology, Department of Microsystems Engineering—IMTEK, University of Freiburg, Freiburg, Germany; [2]Bernstein Center Freiburg, University of Freiburg, Freiburg, Germany

**Abstract** The spatial distribution of neurons and activity-dependent neurite outgrowth shape long-range interaction, recurrent local connectivity and the modularity in neuronal networks. We investigated how this mesoscale architecture develops by interaction of neurite outgrowth, cell migration and activity in cultured networks of rat cortical neurons and show that simple rules can explain variations of network modularity. In contrast to theoretical studies on activity-dependent outgrowth but consistent with predictions for modular networks, spontaneous activity and the rate of synchronized bursts increased with clustering, whereas peak firing rates in bursts increased in highly interconnected homogeneous networks. As $Ca^{2+}$ influx increased exponentially with increasing network recruitment during bursts, its modulation was highly correlated to peak firing rates. During network maturation, long-term estimates of $Ca^{2+}$ influx showed convergence, even for highly different mesoscale architectures, neurite extent, connectivity, modularity and average activity levels, indicating homeostatic regulation towards a common set-point of $Ca^{2+}$ influx.
DOI: https://doi.org/10.7554/eLife.47996.001

**\*For correspondence:**
samora.okujeni@biologie.uni-freiburg.de

**Competing interests:** The authors declare that no competing interests exist.

## Introduction

Modularity is a fundamental design principle of neuronal systems and exists at the scale of cellular compartments, local circuits or interconnected brain areas. From a structural perspective, modularity can arise from inhomogeneities in the physical substrate that facilitate connectivity within a group of functional entities versus connectivity between such groups.

At the mesoscale level of local circuits, the cerebral cortex is organized in local clusters of tightly interconnected neurons (*Feldman and Peters, 1974*; *Skoglund et al., 2004*) that share common inputs and targets (*Bosking et al., 1997*; *Voges et al., 2010*), have similar functional properties (*Ringach et al., 2016*) and are thought to constitute a basic computational module (*Buxhoeveden and Casanova, 2002*; *Casanova and Casanova, 2019*; *Mountcastle, 1997*).

Although cortical architecture is largely genetically predefined at this level, blocking electrical activity during development disturbed the characteristic clustering of connections, suggesting that activity-dependent self-organization influences network modularity (*Durack and Katz, 1996*; *Ruthazer and Stryker, 1996*; *Thompson, 1997*). Intriguingly, computational models predict that modular connectivity, in turn, promotes spontaneous activity (*Kaiser and Hilgetag, 2010*; *Klinshov et al., 2014*; *Mazzucato et al., 2015*). Modularization and spontaneous activity may thus co-evolve in a self-enhancing process.

In early postnatal development, neuronal migration and neurite outgrowth are regulated by activity-dependent changes of the intracellular $Ca^{2+}$ concentration $[Ca^{2+}]_i$ (*Kater and Mills, 1991*; *Komuro and Kumada, 2005*; *Spitzer, 2006*; *Zheng and Poo, 2007*), suggesting that morphodevelopmental processes contribute to cellular $Ca^{2+}$ homeostasis (*Zündorf and Reiser, 2011*). Put

simply, neurons would grow to increase neurite field overlap and the corresponding synaptic connectivity (*Kossio et al., 2018*; *Shepherd et al., 2005*; *Stepanyants et al., 2002*; *Tetzlaff et al., 2010*; *van Ooyen et al., 1995*) to establish the level of spike activity necessary to achieve some target value of $[Ca^{2+}]_i$. As inter-neuron distance strongly affects the overlap of neurite fields and thus connectivity (*Barral and D Reyes, 2016*; *Schnepel et al., 2015*; *Seeman et al., 2018*), spatial clustering of neurons may play an important role in shaping modularity (*Hernández-Navarro et al., 2017*).

In the current study, we focus on the developmental self-organization that leads to different network architectures. In a simple computational model, varying the ratio of activity-dependent homeostatic growth versus migration was sufficient to modify neuronal clustering, mesoscale network organization, and the degree of modularity. Since controlled manipulation of network architecture and simultaneous activity monitoring is impractical in vivo, we tested this developmental interaction by modifying growth and migration in networks of cortical neurons in cell culture. These networks recapitulate major developmental processes such as cell migration and neurite outgrowth (*Guan et al., 2007*; *van Huizen et al., 1987*; *van Pelt et al., 2004*), develop varying degrees of clustering (*Kriegstein and Dichter, 1983*; *Okujeni et al., 2017*; *Soriano et al., 2008*; *Teller et al., 2014*) and produce a rich repertoire of spontaneous bursting events (SBE) (*Kamioka et al., 1996*; *Okujeni et al., 2017*; *Wagenaar et al., 2006*), similar to the developing cortex (*Golshani et al., 2009*; *Minlebaev et al., 2007*).

On the biochemical level, neuronal morphology is regulated by an interplay between activity-dependent kinases and phosphatases controlling cytoskeletal turnover rates (*Flynn, 2013*; *Quinlan and Halpain, 1996*). A key player herein is PKC, a $Ca^{2+}$-modulated enzyme regulating cell migration (*Itoh et al., 1989*; *Larsson, 2006*) and neurite outgrowth (*Gundlfinger et al., 2003*; *Metzger, 2010*).

Increasing PKC activity in cultured networks amplified cell body clustering and local neurite entanglement at the expense of long-range connections, promoting local burst initiation and average firing rate (AFR) but reducing network recruitment during SBEs (*Okujeni et al., 2017*; *Okujeni and Egert, 2019*). This supports the theoretical predictions for modular networks mentioned above and is consistent with results from clustered networks created by mechanical constraints or modified growth substrates (*Bisio et al., 2014*; *Tibau Martorell et al., 2018*; *Yamamoto et al., 2018*).

Irrespective of network architecture, activity stabilized after approximately 21 days in vitro (DIV), suggesting that the target of homeostatic network development had been achieved. Different AFRs at this stage, however, conflict with previous studies assuming that AFR development reflects the homeostatic regulation of $[Ca^{2+}]_i$ (*Abbott and Rohrkemper, 2007*; *Kossio et al., 2018*; *van Ooyen et al., 1995*). $Ca^{2+}$-influx, however, exponentially increases with membrane depolarization (*Mazzanti et al., 1992*) and thus depends on the temporal structure of spike activity. Our findings suggest that because of this non-linearity and specific differences in network-wide peak firing rates (PFR), long-term average $Ca^{2+}$ influx converges despite different AFRs and connectivity. Migration and neurite growth thus interact in a homeostatic process that defines the mesoscale architecture of neuronal networks.

## Results

The connectivity between neurons depends on the overlap of their neurite fields and on their spatial distribution in the network. Like neurite growth, however, this distribution is dynamic because neurons migrate even in postnatal development. In a recurrent network, the input a neuron receives then depends on its embedding as well as the network's overall connectivity and activity structure. Here, we investigated how activity-dependent neurite growth and migration interact to establish connectivity and activity in neuronal networks.

### Simulating activity-dependent neurite growth and migration

To gain insights into interdependencies between neurite growth and neuronal migration during the activity-dependent network self-organization, we extended a network growth model introduced by *van Ooyen et al. (1995)* that reproduces the outgrowth and subsequent pruning of neurites reported for developing neuronal networks (*van Huizen et al., 1987*; *van Pelt et al., 2004*). Following this, neurons were initially randomly seeded on a torus and their interconnectivity was modeled

as degree of overlap between their circular neurite fields (no distinction was made between axons and dendrites). Input to neurons was calculated as the product of presynaptic firing rates and respective connectivity. A sigmoidal transfer function governed the relation between input-dependent membrane potential depolarization and firing rate (*Figure 1A*). A growth process superimposed onto this framework allowed neurons to adjust their input by growing or shrinking their neurite fields, and thus the overlap with other fields, to establish a defined target firing rate (*Figure 1B,C*). In addition to neurite growth, the final phase of neuronal migration observed in postnatal development is modulated by network activity and thus interacts with the formation of neurite fields and the regulation of connectivity. We therefore extended the original framework of the model by adding activity-dependent migration, where neuron somata migrated in the direction of the strongest input and gradually slowed down as their firing rates converged to the target level (*Figure 1B,D*). In contrast to the bidirectional modulation of neurite fields, neurons were not repelled, however, if the activity level was above target. Prior to the formation of first contacts, migration was determined by erratic movements only. Neurons could thus increase their input by extending neurites and by migration to increase the overlap of neurite fields. The relative contribution of migration in network formation herein depended on its rate in relation to the net rate of neurite extension or pruning.

## Migration and neurite outgrowth shape network architecture

Initially, neurite outgrowth (*Figure 2A*) and migration (*Figure 2B*) did not depend on activity. Once neurite fields began to overlap, directed migration towards areas that provided more input amplified statistical variations in the local cell density and led to clustering, indicated by decreasing clustering index (CI, *Figure 2C*). CI was calculated as the ratio between the average nearest neighbor distance in a network and the expected average nearest neighbor distance for random networks. CI above one indicates grid-like cell body arrangements and CI below one indicates clustering. Increasing clustering promoted connectivity buildup (*Figure 2D*) and thus input to a neuron (*Figure 2E*), which advanced the onset of spontaneous network activity (*Figure 2F*). Migration and clustering of neurons ceased with the steep onset of network activity (*Figure 2B,C,F*). In homogeneous networks, neurite fields had to grow larger than in clustered networks to establish the same degree of overlap and thus connectivity (*Figure 2A,D*). As a result, the size of neurites in mature networks correlated negatively with the degree of neuronal clustering (*Figure 2—figure supplement 1*). Connectivity, input activity and firing rates eventually converged to the same levels for different migration conditions (*Figure 2D–F*).

Varying the rate of migration crucially impacted on the overall architecture of developing networks (*Figure 2I*, *Figure 2—videos 1–4*). Without migration, networks developed the most homogeneous neurite field diameters and neurite coverage (*Figure 2G*). Clustering led to more variable neurite field diameters as more isolated neurons required large fields to receive sufficient input, whereas within dense clusters, strongly overlapping neurite fields remained small.

The evolution of the largest connected subnetwork, that is the giant component, suggested that full network connectivity was established along the same developmental time line, irrespective of the degree of clustering (*Figure 2H*, inset). In clustered networks, however, individual neurons played an important role in bridging subnetworks (*Figure 2I*, arrows in the bottom panel). To quantify the tendency for modularity with different architectures, we calculated the giant component remaining after removing increasing subsets of randomly selected neurons in mature networks (*Figure 2H*). In clustered networks, the giant component shrunk faster with an increasing fraction of neurons removed, demonstrating that individual neurons became critical bottlenecks in connectivity. Increasing activity-dependent migration relative to neurite growth thus increased the modularity (Q, *Figure 2I*) of the network.

## Mesoscale network architecture in vitro

The growth model suggested that spatial clustering of neurons during development could play a crucial role in the formation of network connectivity by influencing the probability of neurites to overlap during outgrowth. We assessed this dependence experimentally by chronic activation or inhibition of PKC ($PKC^+$ and $PKC^-$ respectively), a regulator of neuronal migration, in developing networks of cortical neurons in cell culture. As described previously (*Okujeni et al., 2017*), PKC

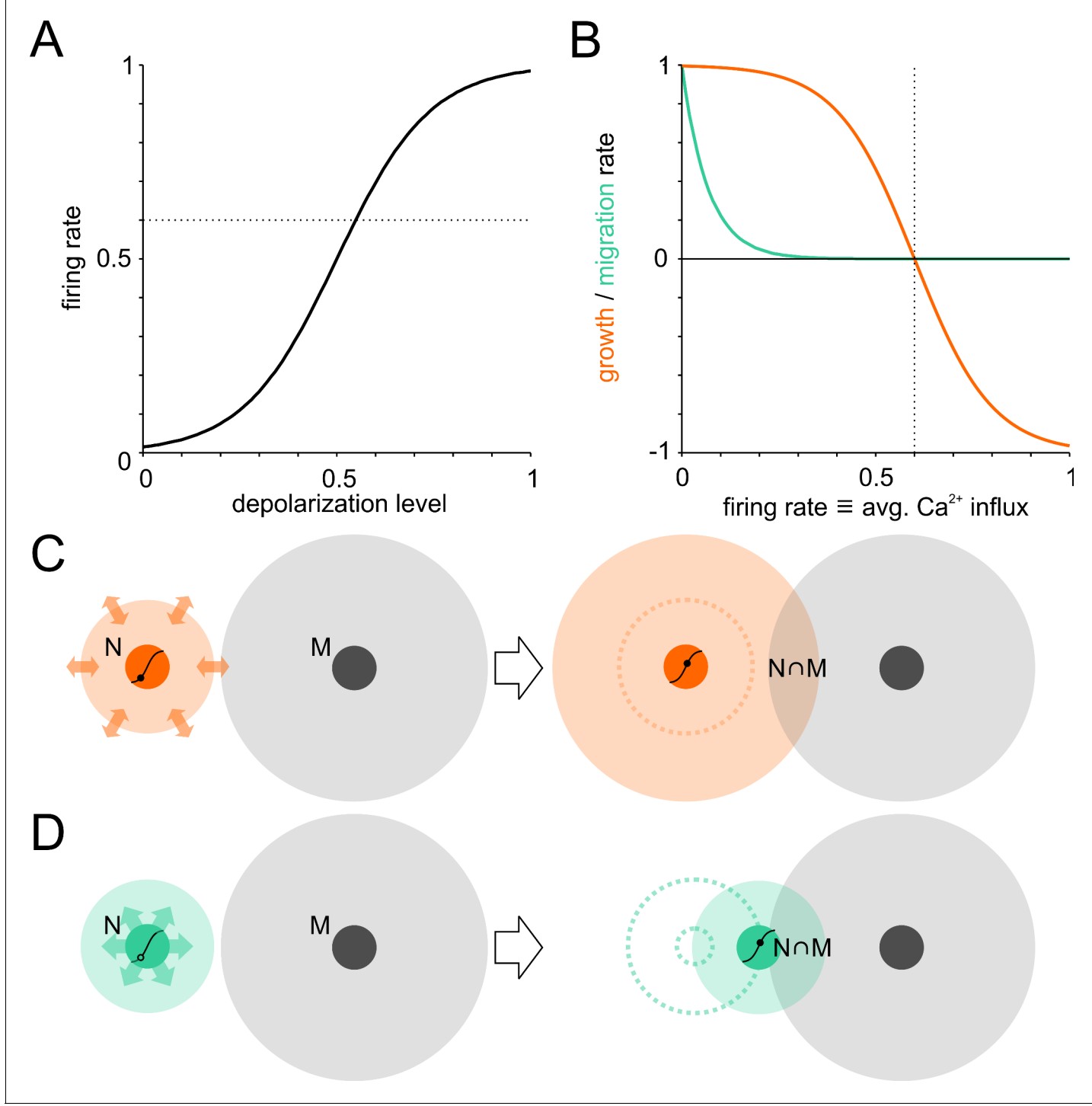

**Figure 1.** Model of activity-dependent network development. Neuronal wiring strategies may involve expansion of neurite fields and migration towards other neurons to increase connectivity modeled as neurite field overlap. (A) Transfer function of membrane depolarization between resting and maximal potential to firing rates. *Dotted line*: target firing rate. (B) Neurite growth (orange) and migration (green) were modulated as a function of $[Ca^{2+}]_i$ that corresponded to average firing rates. Neurites grew while the firing rate (corresponding to long-term average $Ca^{2+}$ influx) was below target and were pruned when above it. Migration rate decreased as neurons approached the target firing rate (*dotted line*). (C) The area of neurite field overlap, corresponding to connectivity in the model, can be increased by neurite outgrowth and neuronal migration towards neighboring neurons (D).
DOI: https://doi.org/10.7554/eLife.47996.002

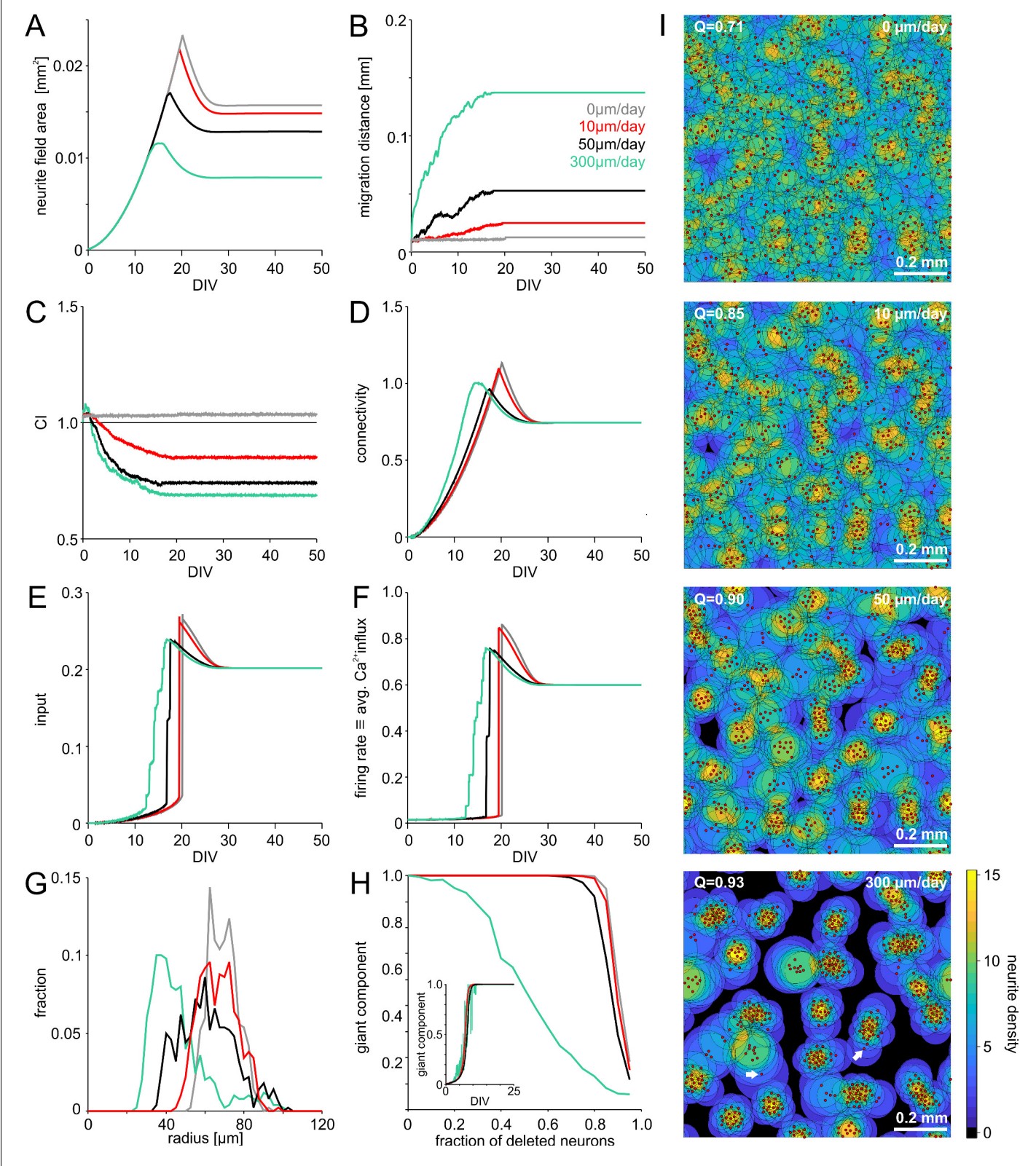

**Figure 2.** Model of activity-dependent growth and migration. (A) Activity-dependent growth produced a characteristic overshoot and subsequent pruning of neurite fields. The overall size of developing neurites decreased with increasing migration rates and clustering. (B) Mean migration distance of neurons after seeding (smoothed by 1 hr sliding average). (C) Migration promoted clustering of neurons, which saturated with the onset of network activity and neurite pruning (curves smoothed by 1 hr sliding average). All networks were initialized with the same spatial cell body distribution with CI

*Figure 2 continued on next page*

*Figure 2 continued*

close to 1. Note that the fluctuations for zero migration results from the random jittering of neuron positions by half the cell body radius (6 µm). (D) Average connectivity increased more rapidly with stronger migration and clustering. (E) Input increased faster with increasing migration rate because clustering initially promoted connectivity. Input levels eventually converged. (F) Firing rates increased sharply once critical input levels were attained. Migration and clustering accelerated the onset of activity. With increasing migration, steps arise because of incremental integration and activation of clusters within the larger network. Note that clustering reduced the developmental overshoot of firing rates. (G) Moderate migration and clustering produced the highest variability of neurite field size across neurons in mature networks. (H) High migration rates increased modularity in mature networks. With increasing migration rate, the giant component more rapidly decreased in clustered networks when a certain fraction of neurons was randomly deleted, indicating that these networks break into disconnected subnets. Inset: the fraction of neurons in the giant cluster, that is the largest connected subnetwork, evolved similarly in different migration conditions. (I) Migration rates crucially determined the mesoscale architecture and modularity (increasing Q indicates stronger modularity) of developing networks. While average neurite fields were small in clustered networks, more isolated neurons generated larger fields (arrows) and formed bottlenecks for activity propagation by connecting otherwise unconnected or weakly connected subnetworks.

DOI: https://doi.org/10.7554/eLife.47996.003

The following video and figure supplements are available for figure 2:

**Figure supplement 1.** Influence of neuronal clustering on neurite field development.
DOI: https://doi.org/10.7554/eLife.47996.004
**Figure supplement 2.** Simulation of saturating network growth.
DOI: https://doi.org/10.7554/eLife.47996.005
**Figure 2—video 1.** Simulated network development with migration rate 0 µm/day.
DOI: https://doi.org/10.7554/eLife.47996.006
**Figure 2—video 2.** Simulated network development with migration rate 10 µm/day.
DOI: https://doi.org/10.7554/eLife.47996.007
**Figure 2—video 3.** Simulated network development with migration rate 50 µm/day.
DOI: https://doi.org/10.7554/eLife.47996.008
**Figure 2—video 4.** Simulated network development with migration rate 300 µm/day.
DOI: https://doi.org/10.7554/eLife.47996.009

manipulation significantly altered the mesoscale architecture of networks with 600–800 neurons/mm$^2$ (*Figure 3A*), with striking similarity to mature networks generated with the growth model. Under control conditions (PKC$^N$ networks), networks appeared as inhomogeneous density landscapes with both, clustered and sparse regions (*Figure 3A*, center panel). In particular in clustered areas, neurites formed tangles, which would increase the probability of local connections. Axons spanning several millimeters indicated monosynaptic connections between distant network regions. In comparison, PKC$^-$ networks with diminished migration had a more homogeneous distribution of cell bodies and coverage with dendrites and axons (*Figure 3A*, left panel). Reduced fasciculation of neurites and a high density of long-range axons suggested a more isotropic embedding of neurons and more random-like connectivity. In turn, PKC$^+$ networks with enhanced migration had well delineated clusters of about 30–60 neurons with dense tangles of neurites that rarely reached into neighboring clusters (*Figure 3A*, right panel), indicating high local connectivity and reduced inter-cluster connectivity.

## Cell migration promotes neuronal clustering

To quantify the structural development, we seeded networks at lower densities of about 300 neurons per mm$^2$ that were more suitable for morphometric analyses (*Figure 3—figure supplement 1*). Within the first day of random seeding of neurons, rapid neurite outgrowth resulted in overlapping neurite fields between neighboring neurons. Simultaneously, neuronal cell bodies migrated across the substrate. Neuronal migration with concurrent outgrowth of neurites gradually increased neuron clustering within about three weeks in vitro (*Figure 3B*). Chronic manipulation of PKC activity differentially modulated neuronal clustering during development (*Table 1*). At 22 DIV, clustering was moderate in PKC$^N$ networks (CI = 0.75 ± 0.03) but significantly increased in the PKC$^+$ networks (CI = 0.67 ± 0.02, p=3.3*10$^{-2}$) and significantly reduced in the PKC$^-$ networks (CI = 0.88 ± 0.01, p=4.4*10$^{-4}$). CI did not change significantly after 22 DIV, indicating cessation of neuronal migration.

Note that the spatial patterning of somata depended on neuron density. Clusters in dense networks (~700 neurons/mm$^2$ at >22 DIV) typically contained 30–60 neurons (*Okujeni et al., 2017*),

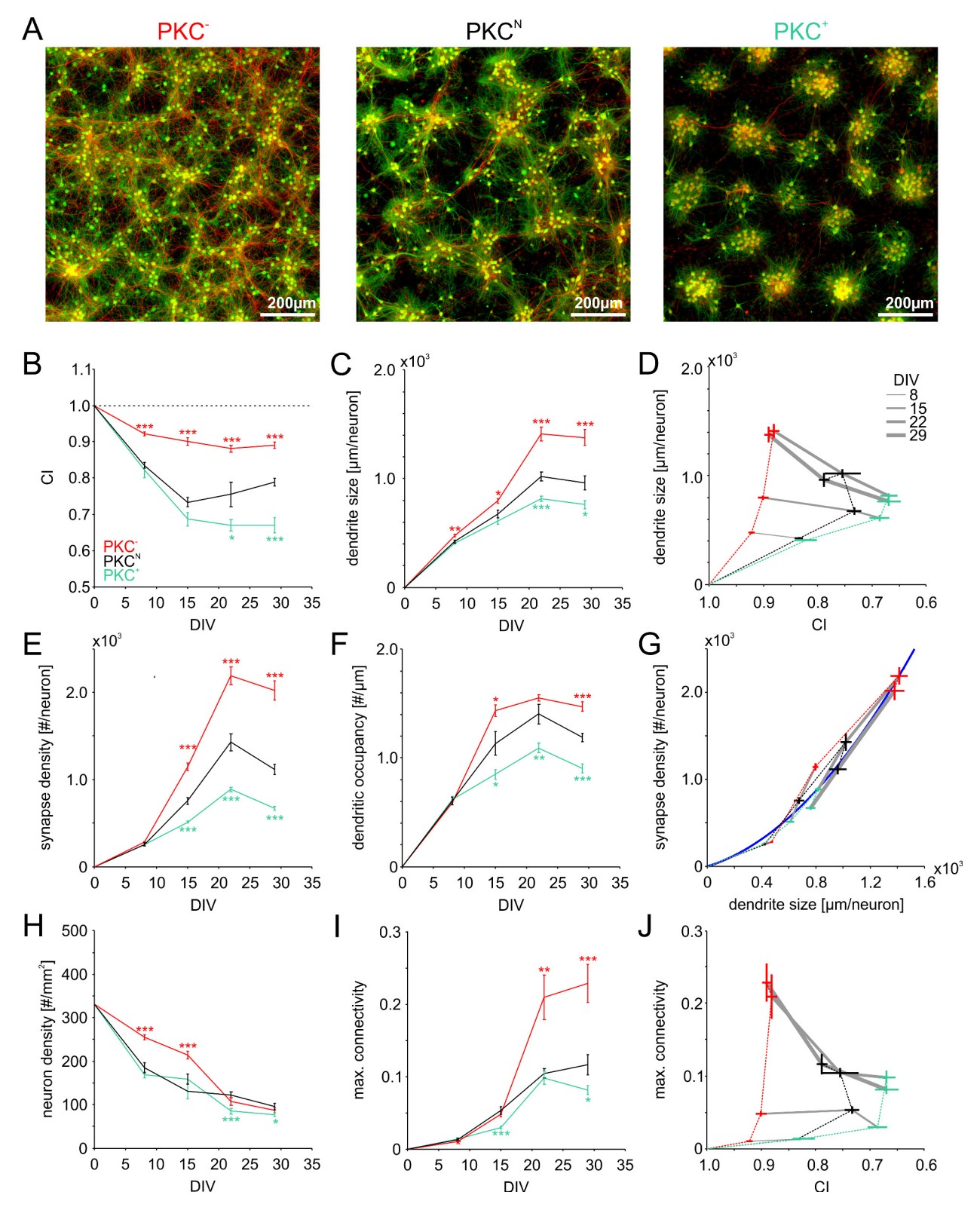

**Figure 3.** Morphometric analyses of network development. (**A**) Dense networks established characteristic mesoscale architectures for the different PKC conditions. PKC⁻ networks had a more homogeneous distribution of axons (red), dendrites (green) and cell bodies (green) than PKCᴺ networks. In PKC⁺ networks, neurons formed well-delineated clusters. Note that the following morphometric analyses are based on sparser cultures. (**B**) Decreasing CI during development reflects cell migration and ongoing clustering of neuronal cell bodies until ~15 DIV. PKC⁺ promoted and PKC⁻ diminished

*Figure 3 continued on next page*

*Figure 3 continued*

clustering during development. (**C**) Dendrite size increased until 22 DIV, with boosted growth in PKC⁻ and diminished growth in PKC⁺ networks. (**D**) In late development, dendrite size scaled inversely with the degree of clustering. For visualization the CI axis was inversed, so the degree of clustering increases from left to right. (**E**) The synapse density increased concurrently with dendrite growth. After 22 DIV synapse densities decreased in PKC$^N$ and PKC⁺ networks, indicating synaptic pruning. (**F**) Dendritic occupancy with synapses differed slightly between conditions and decreased after 22 DIV. (**G**) The number of synapse per neurons increased with the dendrite size. Gray lines connect networks of the same age. The blue line illustrates a proposed quadratic scaling rule between dendrite size and synapse densities. (**H**) Neuron density declined with DIV to about one third of the seeding density. (**I**) Estimated upper bounds for connectivity based on the synapse density and the total number of neurons (on 113 mm² cover slips). PKC⁻ at least doubled average connectivity. (**J**) In mature networks, maximum connectivity scaled inversely with clustering. All parameters are presented as mean ± SEM. Data from 4 to 24 images (*Table 1*, area 3.5 mm²) taken in each of 2 networks per condition and age. Asterisks indicate p-values ≤0.05 (*), ≤0.01 (**) and ≤0.001 (***) tested against PKC$^N$.

DOI: https://doi.org/10.7554/eLife.47996.010

The following source data and figure supplements are available for figure 3:

**Source data 1.** Source data and Matlab script for *Figure 3B,C,E,F,H,I*.
DOI: https://doi.org/10.7554/eLife.47996.013
**Figure supplement 1.** Clustering and dendrite development in sparse networks.
DOI: https://doi.org/10.7554/eLife.47996.011
**Figure supplement 2.** Development of synapses in sparse networks.
DOI: https://doi.org/10.7554/eLife.47996.012

whereas in sparse networks (~100 neurons/mm² at >22 DIV) clusters mostly consisted of fewer than 10 neurons (*Figure 3—figure supplement 1*).

## Clustering diminishes dendrite outgrowth

To address the interaction of neurite field extension, migration and clustering, we analyzed the average size of dendrites at several time points during development (*Figure 3C*). Dendrite size was quantified as the ratio between the total length of detected dendrite stretches and the number of neurons within regions of interest (*Table 1*). The measure estimates the average contribution of each neuron to the dendritic mesh. Chronic manipulation of PKC activity had little impact on dendrite size up to 8 DIV but significantly modulated dendrite outgrowth during subsequent development. At 22 DIV, dendrite size was significantly increased in the more homogeneous PKC⁻ networks but significantly reduced in the more strongly clustered PKC⁺ networks (PKC$^N$: 1021 ± 41 μm; PKC⁻: 1413 ± 64 μm, p=7.9*10⁻⁵; PKC⁺: 816 ± 24 μm, p=3.6*10⁻⁴). In all conditions, dendrite size did not change significantly between 22 and 29 DIV, indicating stabilization of the dendritic network after the third week in vitro. As in the model (*Figure 2—figure supplement 1*), dendrite size in mature networks was negatively correlated with the degree of cell body clustering and, thus, the distance between neurons (*Figure 3D*).

## Dendrite outgrowth promotes synaptic connectivity

Network connectivity requires neurite overlap but further depends on the probability by which synapses are realized at axo-dendritic intersections. To assess how synaptic connectivity evolved in the different PKC conditions, we stained and detected presynaptic boutons (*Figure 3—figure supplement 2*) and determined the synapse density as the average number of presynaptic boutons per neuron (*Figure 3E*, *Table 1*) and the dendritic occupancy as the number of synapses per unit dendrite length (*Figure 3F*, *Table 1*). Manipulating PKC activity had no significant influence on early synaptogenesis up to 8 DIV, consistent with the comparable dendrite density in different PKC conditions at this stage. Paralleling dendritic outgrowth, synapse density increased significantly with increasing dendritic occupancy between 8 and 22 DIV in all conditions. Synapse densities and dendritic occupancy subsequently decreased between 22 and 29 DIV. This reduction was not significant in PKC⁻ networks, however. Developmental manipulation of PKC activity profoundly affected mature synapse densities (PKC$^N$: 1114 ± 56; PKC⁻: 2019 ± 110, p=4.5*10⁻⁸; PKC⁺: 669 ± 21, p=5.7*10⁻⁸) and dendritic occupancy (PKC$^N$: 1.19 ± 0.04 μm⁻¹; PKC⁻: 1.47 ± 0.04, p=3.0*10⁻⁵ μm⁻¹; PKC⁺: 0.9 ± 0.04 μm⁻¹, p=2.3*10⁻⁵) at 29 DIV, both of which were significantly increased in the PKC⁻ and reduced in the PKC⁺ condition. Similar to dendrite densities, synapse densities were thus negatively correlated with the degree of clustering. Across PKC conditions and developmental stages, synapse

**Table 1.** Morphometric analysis of network development under different PKC conditions.

Results are presented as mean ± standard error of mean (SEM). Significance was determined against PKC$^N$, or between specified developmental time windows, using independent Student's t-test. *N* specifies the number of analyzed images taken from two networks per PKC condition and age.

| | DIV | PKC$^-$ | PKC$^N$ | PKC$^+$ | unit |
|---|---|---|---|---|---|
| **clustering index** | | | | | |
| | 8 | 0.92 ± 0.01 ($1.8*10^{-10}$) | 0.84 ± 0.01 | 0.82 ± 0.02 ($6.1*10^{-1}$) | CI |
| | 15 | 0.9 ± 0.01 ($1.5*10^{-6}$) | 0.73 ± 0.01 | 0.69 ± 0.02 ($1.1*10^{-1}$) | CI |
| | 22 | 0.88 ± 0.01 ($4.4*10^{-4}$) | 0.75 ± 0.03 | 0.67 ± 0.02 ($3.3*10^{-2}$) | CI |
| | 29 | 0.89 ± 0.01 ($8.0*10^{-9}$) | 0.79 ± 0.01 | 0.67 ± 0.02 ($2.1*10^{-5}$) | CI |
| | 8 vs. 22 | −4.49 ($3.5*10^{-4}$) | −9.65 ($3.1*10^{-2}$) | −18.6 ($1.2*10^{-5}$) | % change |
| | 22 vs. 29 | 1.04 ($4.9*10^{-1}$) | 4.5 ($2.7*10^{-1}$) | 0.01 (1.0) | % change |
| **Dendrite size** | | | | | |
| | 8 | 476 ± 9 ($1.3*10^{-3}$) | 421 ± 12 | 408 ± 8 ($3.4*10^{-1}$) | µm |
| | 15 | 797 ± 22 ($1.2*10^{-2}$) | 676 ± 37 | 610 ± 30 ($2.1*10^{-1}$) | µm |
| | 22 | 1413 ± 64 ($7.9*10^{-5}$) | 1021 ± 41 | 816 ± 24 ($3.6*10^{-4}$) | µm |
| | 29 | 1380 ± 74 ($1.9*10^{-4}$) | 962 ± 65 | 760 ± 37 ($1.3*10^{-2}$) | µm |
| | 8 vs. 22 | 196.59 ($4.0*10^{-20}$) | 142.46 ($9.1*10^{-12}$) | 100.08 ($1.4*10^{-15}$) | % change |
| | 22 vs. 29 | −2.38 ($7.4*10^{-1}$) | −5.81 ($5.0*10^{-1}$) | −6.86 ($2.5*10^{-1}$) | % change |
| **Synapse density** | | | | | |
| | 8 | 281 ± 11 ($2.2*10^{-1}$) | 255 ± 18 | 254 ± 14 ($9.4*10^{-1}$) | # |
| | 15 | 1142 ± 44 ($2.3*10^{-4}$) | 754 ± 39 | 510 ± 9 ($2.4*10^{-5}$) | # |
| | 22 | 2188 ± 100 ($2.1*10^{-5}$) | 1427 ± 99 | 885 ± 27 ($3.4*10^{-5}$) | # |
| | 29 | 2019 ± 110 ($4.5*10^{-8}$) | 1114 ± 56 | 669 ± 21 ($5.7*10^{-8}$) | # |
| | 8 vs. 22 | 678.77 ($4.7*10^{-24}$) | 458.69 ($2.1*10^{-10}$) | 248.68 ($1.2*10^{-17}$) | % change |
| | 22 vs. 29 | −7.75 ($2.7*10^{-1}$) | −21.93 ($6.7*10^{-3}$) | −24.35 ($1.2*10^{-6}$) | % change |
| **Dendritic occupancy** | | | | | |
| | 8 | 0.59 ± 0.02 ($6.4*10^{-1}$) | 0.6 ± 0.04 | 0.62 ± 0.03 ($7.5*10^{-1}$) | #/µm |
| | 15 | 1.44 ± 0.05 ($1.7*10^{-2}$) | 1.13 ± 0.11 | 0.85 ± 0.04 ($1.7*10^{-2}$) | #/µm |
| | 22 | 1.55 ± 0.03 ($9.6*10^{-2}$) | 1.4 ± 0.09 | 1.09 ± 0.04 ($5.7*10^{-3}$) | #/µm |
| | 29 | 1.47 ± 0.04 ($3.0*10^{-5}$) | 1.19 ± 0.04 | 0.9 ± 0.04 ($2.3*10^{-5}$) | #/µm |
| | 8 vs. 22 | 163.83 ($7.2*10^{-28}$) | 132.26 ($9.0*10^{-8}$) | 76.65 ($3.7*10^{-10}$) | % change |
| | 22 vs. 29 | −5.06 ($1.5*10^{-1}$) | −15.41 ($2.3*10^{-2}$) | −17.45 ($3.8*10^{-3}$) | % change |
| **Neuron density** | | | | | |
| | 8 | 255 ± 6 ($9.6*10^{-7}$) | 185 ± 11 | 168 ± 7 ($2.0*10^{-1}$) | #/mm$^2$ |
| | 15 | 214 ± 9 ($5.9*10^{-4}$) | 131 ± 17 | 158 ± 12 ($2.0*10^{-1}$) | #/mm$^2$ |
| | 22 | 107 ± 8 ($1.9*10^{-1}$) | 123 ± 6 | 85 ± 6 ($5.7*10^{-4}$) | #/mm$^2$ |
| | 29 | 87 ± 5 ($3.0*10^{-1}$) | 96 ± 7 | 77 ± 4 ($2.6*10^{-2}$) | #/mm$^2$ |
| | 8 vs. 22 | −58.03 ($7.3*10^{-17}$) | −33.66 ($1.1*10^{-4}$) | −49.42 ($1.0*10^{-8}$) | % change |
| | 22 vs. 29 | −18.93 ($4.5*10^{-2}$) | −21.71 ($1.3*10^{-2}$) | −9.79 ($2.6*10^{-1}$) | % change |
| **Maximum connectivity** | | | | | |
| | 8 | 0.01 ± 0.001 ($3.5*10^{-2}$) | 0.013 ± 0.001 | 0.014 ± 0.001 ($6.3*10^{-1}$) | fraction |
| | 15 | 0.048 ± 0.003 ($4.6*10^{-1}$) | 0.053 ± 0.006 | 0.029 ± 0.002 ($9.2*10^{-4}$) | fraction |
| | 22 | 0.209 ± 0.031 ($8.2*10^{-3}$) | 0.104 ± 0.007 | 0.098 ± 0.009 ($5.9*10^{-1}$) | fraction |
| | 29 | 0.229 ± 0.026 ($9.2*10^{-4}$) | 0.116 ± 0.014 | 0.081 ± 0.006 ($3.8*10^{-2}$) | fraction |
| | 8 vs. 22 | 1987.43 ($5.9*10^{-10}$) | 701.18 ($2.6*10^{-11}$) | 604.32 ($1.4*10^{-10}$) | % change |

*Table 1 continued on next page*

*Table 1 continued*

| DIV | PKC⁻ | PKCᴺ | PKC⁺ | unit |
|---|---|---|---|---|
| 22 vs. 29 | 9.31 (6.3*10⁻¹) | 11.48 (5.2*10⁻¹) | −17.23 (1.3*10⁻¹) | % change |

| N | | | | |
|---|---|---|---|---|
| 8 | 24 | 11 | 15 | |
| 15 | 9 | 4 | 7 | |
| 22 | 15 | 11 | 11 | |
| 29 | 17 | 16 | 15 | |

DOI: https://doi.org/10.7554/eLife.47996.014
The following source data is available for Table 1:
**Source data 1.** Source data and Matlab script.
DOI: https://doi.org/10.7554/eLife.47996.015
**Source data 2.** Source data and Matlab script.
DOI: https://doi.org/10.7554/eLife.47996.016

occupancy scaled approximately quadratic with the dendrite size (*Figure 3G*), which could result from similarly modulated axonal densities (*Okujeni et al., 2017*) and the corresponding multiplicative increase in intersection probability.

## Clustering reduces maximum global connectivity

Network connectivity is limited by the number of synapses per neuron and the overall number of neurons in a network since neurons obviously cannot have more partners than they have synapses. The ratio between the number of synapses per neuron and the total number of neurons in the network defines an upper bound of connectivity for a network (maximum connectivity). The degree of connectivity realized, however, could be lower because of multiple structural synapses between neuron pairs. Although the density of neurons decreased during early development (*Figure 3H*, *Table 1*), maximum connectivity increased significantly in all conditions between 8 and 22 DIV (*Figure 3I*) and saturated between 22–29 DIV. At the same time, maximum connectivity almost doubled in PKC⁻ networks compared to PKCᴺ networks but was significantly reduced in PKC⁺ networks (PKCᴺ: $0.12 \pm 0.01$; PKC⁻: $0.23 \pm 0.03$, $p=9.2*10^{-4}$; PKC⁺: $0.08 \pm 0.01$, $p=3.8*10^{-2}$) and thus was negatively correlated with the degree of clustering (*Figure 3J*).

## Mesoscale architecture and the development of spontaneous activity

We recently showed that the specific spatiotemporal patterns of spontaneous bursting depended considerably on the mesoscale architecture of the network (*Okujeni et al., 2017*; *Okujeni and Egert, 2019*) (*Figure 4—figure supplement 1*). In all networks types, spikes were typically organized in bursts that were synchronized across micro-electrode arrays (MEA; *Figure 4—figure supplement 1A,B*) with low activity between SBEs. Mature PKC⁻ networks typically generated strong SBEs at low rates with many spikes per recording site (*Figure 4—figure supplement 1C*). SBE rates were significantly increased in moderately clustered PKCᴺ networks with fewer spikes per site (*Figure 4—figure supplement 1D*). Strongly clustered PKC⁺ networks generated weaker SBEs at even higher rates and fewer participating sites (*Figure 4—figure supplement 1E*).

During development, spontaneous activity started with sporadic, uncorrelated spiking in all networks. First SBEs typically appeared at very low rates at 3–5 DIV, indicating that neuronal migration and neurite outgrowth had connected neurons sufficiently to synchronize their activity. Subsequently, activity became increasingly dominated by SBEs attaining mature levels with around 70–90% of all spikes in SBEs at 10–14 DIV (PKCᴺ: $82 \pm 2\%$; PKC⁻: $87 \pm 2\%$; PKC⁺: $71 \pm 3\%$). SBE rates increased faster with enhanced migration and clustering (*Figure 4A*, *Table 2*). The positive correlation between the degree of clustering and SBE rates persisted beyond the end of the migratory phase (10–14 DIV) where SBE rates continued to increase in all PKC conditions until stabilizing after the fourth week. In late development (28–35 DIV), SBE rates were significantly increased in the clustered PKC⁺ networks and reduced in the more homogeneous PKC⁻ networks (PKCᴺ: $17.0 \pm 1.1$ min⁻¹; PKC⁻: $5.0 \pm 0.8$ min⁻¹, $p=2.2*10^{-13}$; PKC⁺: $41.1 \pm 5.1$ min⁻¹, $p=7.9*10^{-9}$).

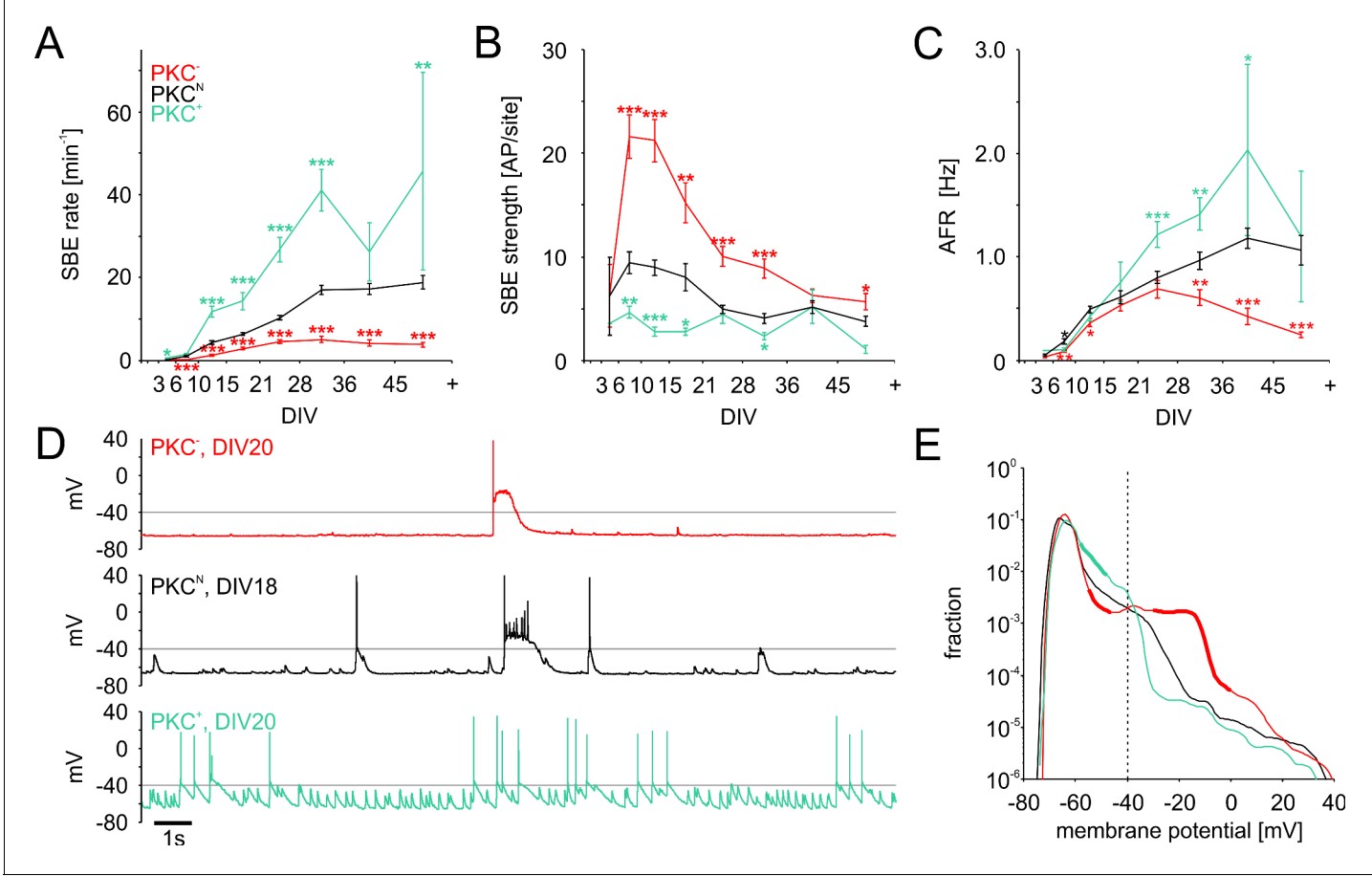

**Figure 4.** Development of spontaneous network activity. (A) SBE rates gradually increased during development until 28 DIV, which was accelerated in clustered PKC$^+$ networks and decelerated in homogeneous PKC$^-$ networks. In result, SBE rates differed considerably in mature networks and increased with the degree of clustering. X-axis ticks indicate bin boundaries. (B) PKC$^-$ networks generated stronger SBEs with more APs per site than clustered networks, compensating lower SBE rates to some extent. Burst strength increased initially but declined later on, putatively because of the maturation of inhibition. (C) AFR increased comparably during early development in the different PKC conditions, indicating that stronger bursting compensated lower SBE rates. Later in development, AFRs were increased in PKC$^+$ and reduced in PKC$^-$ networks. (D) Neurons in PKC$^-$ networks showed strong depolarization during SBEs that reached well above the spiking threshold around −40 mV causing depolarization block of spiking. In clustered networks, neurons displayed higher membrane potential fluctuations below threshold that occasionaly passed the threshold leading to spikes. (E) Membrane potential distribution. Thick lines indicate regions significantly different from PKC$^N$ (p≤0.05). The fraction of time in which neuronal membrane potentials were above the spiking threshold (dashed line) was significantly increased in PKC$^-$ networks compared to PKC$^N$ and PKC$^+$ networks. Data in A-D and F show mean ± SEM derived from 1 hr recording sessions. Asterisks indicate p-values ≤0.05 (*), ≤0.01 (**) and ≤0.001 (***) tested against PKC$^N$. The number of recordings per age and condition is provided in **Table 2**.

DOI: https://doi.org/10.7554/eLife.47996.017

The following source data and figure supplement are available for figure 4:

**Source data 1.** Source data and Matlab script for **Figure 4A–C,E**.
DOI: https://doi.org/10.7554/eLife.47996.019
**Figure supplement 1.** Sample MEA recordings from dense networks.
DOI: https://doi.org/10.7554/eLife.47996.018

Clustering thus promoted spontaneous activity generation, in line with predictions from simulations (**Kaiser and Hilgetag, 2010**) but inconsistent with homeostatic regulation of connectivity towards a target firing rate. Achieving defined AFR by homeostasis would require that increased SBE rates be counterbalanced by a proportional reduction in SBE strength, that is the average number of spikes per SBE. In early development, SBE strength rapidly increased and plateaued at levels that were indeed inversely correlated to SBE rates (**Figure 4B**, **Table 2**), which resulted in similar AFRs across PKC conditions at this age (**Figure 4C**, **Table 2**). Later in development, however, the decline in SBE

**Table 2.** Electrophysiological characterization of network activity during development.
Data were pooled within defined developmental time windows. Significance was determined against
$PKC^N$ using independent Student's t-test. $N$ specifies the number of recorded networks per PKC condition and age.

| | DIV | PKC$^-$ | PKC$^N$ | PKC$^+$ |
|---|---|---|---|---|
| AFR | | | | |
| | 3–5 | $0.03 \pm 0.01$ ($1.0*10^{-1}$) | $0.05 \pm 0.01$ | $0.09$ ($1.4*10^{-1}$) |
| | 6–9 | $0.08 \pm 0.01$ ($2.6*10^{-3}$) | $0.18 \pm 0.03$ | $0.1 \pm 0.02$ ($4.0*10^{-2}$) |
| | 10–14 | $0.36 \pm 0.04$ ($1.8*10^{-2}$) | $0.49 \pm 0.03$ | $0.42 \pm 0.04$ ($2.0*10^{-1}$) |
| | 15–20 | $0.53 \pm 0.06$ ($3.6*10^{-1}$) | $0.61 \pm 0.06$ | $0.76 \pm 0.19$ ($3.6*10^{-1}$) |
| | 21–27 | $0.69 \pm 0.08$ ($2.5*10^{-1}$) | $0.8 \pm 0.06$ | $1.21 \pm 0.12$ ($7.3*10^{-4}$) |
| | 28–35 | $0.6 \pm 0.07$ ($2.9*10^{-3}$) | $0.96 \pm 0.09$ | $1.41 \pm 0.15$ ($6.9*10^{-3}$) |
| | 36–44 | $0.42 \pm 0.08$ ($2.4*10^{-7}$) | $1.18 \pm 0.1$ | $2.04 \pm 0.83$ ($4.3*10^{-2}$) |
| | 45+ | $0.24 \pm 0.02$ ($9.9*10^{-8}$) | $1.06 \pm 0.14$ | $1.2 \pm 0.63$ ($8.3*10^{-1}$) |
| SBE rate (SBE/min) | | | | |
| | 3–5 | $0.17 \pm 0.04$ ($2.9*10^{-1}$) | $0.11 \pm 0.03$ | $0.54$ ($1.5*10^{-2}$) |
| | 6–9 | $0.18 \pm 0.03$ ($5.4*10^{-6}$) | $1.02 \pm 0.15$ | $1.46 \pm 0.19$ ($7.4*10^{-2}$) |
| | 10–14 | $1.21 \pm 0.13$ ($2.0*10^{-8}$) | $4.26 \pm 0.44$ | $11.69 \pm 1.27$ ($3.4*10^{-10}$) |
| | 15–20 | $2.83 \pm 0.35$ ($1.4*10^{-8}$) | $6.36 \pm 0.43$ | $14.31 \pm 2.03$ ($4.7*10^{-7}$) |
| | 21–27 | $4.58 \pm 0.44$ ($2.4*10^{-10}$) | $10.3 \pm 0.62$ | $26.82 \pm 3$ ($5.5*10^{-12}$) |
| | 28–35 | $4.98 \pm 0.81$ ($2.2*10^{-13}$) | $16.97 \pm 1.07$ | $41.1 \pm 5.07$ ($7.9*10^{-9}$) |
| | 36–44 | $4.08 \pm 0.75$ ($9.0*10^{-12}$) | $17.25 \pm 1.28$ | $26.21 \pm 7.05$ ($8.7*10^{-2}$) |
| | 45+ | $3.74 \pm 0.56$ ($2.2*10^{-13}$) | $18.85 \pm 1.62$ | $45.76 \pm 23.86$ ($2.0*10^{-3}$) |
| SBE strength (APs per burst) | | | | |
| | 3–5 | $6.2 \pm 3$ ($1.0*10^{0}$) | $6.2 \pm 3.8$ | $3.5$ ($7.6*10^{-1}$) |
| | 6–9 | $21.6 \pm 2.1$ ($9.4*10^{-7}$) | $9.5 \pm 1$ | $4.6 \pm 0.6$ ($1.2*10^{-3}$) |
| | 10–14 | $21.2 \pm 2.1$ ($5.7*10^{-9}$) | $9 \pm 0.8$ | $2.7 \pm 0.5$ ($1.5*10^{-7}$) |
| | 15–20 | $15.2 \pm 1.9$ ($2.1*10^{-3}$) | $8 \pm 1.3$ | $2.7 \pm 0.3$ ($1.2*10^{-2}$) |
| | 21–27 | $10.1 \pm 1$ ($6.1*10^{-8}$) | $4.9 \pm 0.4$ | $4.4 \pm 0.9$ ($5.4*10^{-1}$) |
| | 28–35 | $8.9 \pm 0.9$ ($2.9*10^{-6}$) | $4 \pm 0.5$ | $2.3 \pm 0.3$ ($1.7*10^{-2}$) |
| | 36–44 | $6.2 \pm 0.6$ ($2.2*10^{-1}$) | $5.1 \pm 0.6$ | $5.1 \pm 1.6$ ($9.8*10^{-1}$) |
| | 45+ | $5.6 \pm 0.8$ ($4.2*10^{-2}$) | $3.8 \pm 0.5$ | $1.1 \pm 0.4$ ($2.2*10^{-1}$) |
| PFR (Hz) | | | | |
| | 3–5 | $12.3 \pm 3.8$ ($9.0*10^{-1}$) | $13.2 \pm 7.3$ | $11.5$ ($9.2*10^{-1}$) |
| | 6–9 | $50.8 \pm 4.8$ ($6.4*10^{-5}$) | $28.3 \pm 2.7$ | $17.4 \pm 1.7$ ($4.9*10^{-3}$) |
| | 10–14 | $76.6 \pm 5.4$ ($6.7*10^{-14}$) | $32.1 \pm 2.3$ | $10.3 \pm 1.4$ ($2.1*10^{-9}$) |
| | 15–20 | $59.1 \pm 6.5$ ($5.3*10^{-5}$) | $29.8 \pm 3.4$ | $10.9 \pm 1.3$ ($6.4*10^{-4}$) |
| | 21–27 | $43.3 \pm 4.1$ ($7.4*10^{-10}$) | $18.9 \pm 1.5$ | $13.7 \pm 2$ ($4.4*10^{-2}$) |
| | 28–35 | $42.3 \pm 4.4$ ($2.4*10^{-8}$) | $15.5 \pm 2$ | $8.1 \pm 1.2$ ($1.8*10^{-2}$) |
| | 36–44 | $30.5 \pm 3.1$ ($2.7*10^{-2}$) | $21.4 \pm 2.6$ | $6.1 \pm 0.4$ ($1.3*10^{-1}$) |
| | 45+ | $27.5 \pm 3.8$ ($1.5*10^{-2}$) | $16.4 \pm 2.3$ | $5.6 \pm 1.7$ ($2.8*10^{-1}$) |
| Network synchrony | | | | |
| | 3–5 | $0.1 \pm 0.03$ ($2.6*10^{-1}$) | $0.04 \pm 0.02$ | $0.08$ ($3.4*10^{-1}$) |
| | 6–9 | $0.39 \pm 0.02$ ($3.3*10^{-3}$) | $0.29 \pm 0.02$ | $0.15 \pm 0.02$ ($3.1*10^{-4}$) |
| | 10–14 | $0.52 \pm 0.03$ ($2.5*10^{-10}$) | $0.31 \pm 0.02$ | $0.12 \pm 0.02$ ($1.4*10^{-10}$) |
| | 15–20 | $0.53 \pm 0.04$ ($4.6*10^{-5}$) | $0.35 \pm 0.02$ | $0.16 \pm 0.03$ ($1.7*10^{-5}$) |

*Table 2 continued on next page*

*Table 2 continued*

| DIV | PKC$^-$ | PKC$^N$ | PKC$^+$ |
|---|---|---|---|
| 21–27 | 0.51 ± 0.03 (5.8*10$^{-13}$) | 0.26 ± 0.02 | 0.2 ± 0.02 (4.8*10$^{-2}$) |
| 28–35 | 0.57 ± 0.04 (2.0*10$^{-10}$) | 0.24 ± 0.03 | 0.15 ± 0.03 (7.6*10$^{-2}$) |
| 36–44 | 0.53 ± 0.04 (1.3*10$^{-5}$) | 0.3 ± 0.03 | 0.11 ± 0.03 (1.3*10$^{-1}$) |
| 45+ | 0.45 ± 0.05 (2.5*10$^{-3}$) | 0.26 ± 0.03 | 0.14 ± 0.11 (3.8*10$^{-1}$) |
| N | | | |
| 3–5 | 7 | 3 | 1 |
| 6–9 | 33 | 40 | 24 |
| 10–14 | 70 | 92 | 47 |
| 15–20 | 53 | 65 | 27 |
| 21–27 | 77 | 121 | 56 |
| 28–35 | 47 | 62 | 29 |
| 36–44 | 38 | 57 | 4 |
| 45+ | 38 | 36 | 2 |

DOI: https://doi.org/10.7554/eLife.47996.020

strength was not proportional to the increase in SBE rates, in particular in PKC$^-$ networks. This resulted in significantly lower AFRs in PKC$^-$ networks (0.6 ± 0.1 Hz, p=2.9*10$^{-3}$) and significantly increased AFRs in PKC$^+$ networks (1.4 ± 0.2 Hz, p=6.9*10$^{-3}$) compared to PKC$^N$ networks (1.0 ± 0.1 Hz) at 28–35 DIV.

## Clustering decreases PFR and depolarization during SBEs

The hypothetical set-point of the homeostatic process, however, is not the firing rate per se but the associated $[Ca^{2+}]_i$ (*Mattson and Kater, 1987*), which is linked to molecular processes involved in growth and migration. $Ca^{2+}$ influx increases supra-linearly with increasing membrane depolarization (*Mazzanti et al., 1992*). This suggests that the long-term $Ca^{2+}$ gain is not a linear function of AFR but depends the depolarization of the membrane potential and thus on the temporal structure of activity. Depolarization depends on the number and synchronization of excitatory synaptic input, which becomes maximal during the peak phase of SBEs. Simultaneous intracellular and extracellular recording showed that higher SBE strength was indeed associated with stronger depolarization during SBEs (*Okujeni et al., 2017*). In PKC$^-$ networks, membrane depolarization high above spiking threshold frequently led to a depolarization block that outlasted the spike burst (*Figure 4D* top trace). The fraction of time spent above threshold (−40 mV, *Figure 4E*) was significantly larger in neurons of PKC$^-$ networks (5.2 ± 0.7%, p=1.7*10$^{-4}$, N = 30 neurons; mean ± SEM, independent Student's t-test) than in PKC$^N$(1.7 ± 0.5%, N = 24) and PKC$^+$(1.2 ± 0.7%, p=1.2*10$^{-3}$, N = 24) networks (14–23 DIV). Depolarization was therefore not necessarily correlated with the individual firing rate of a neuron and the AFR in the network but rather reflected the network PFR during SBEs.

## Homeostatic regulation of growth by long-term $Ca^{2+}$ influx

To assess how $Ca^{2+}$ influx depends on PFR, we determined the amplitude of $Ca^{2+}$ transients in excitatory neurons expressing GCaMP under the CAMKII promotor while simultaneously recording SBEs with MEAs (*Figure 5A*). Most neurons indeed showed an exponential relation between PFR and the amplitude of $Ca^{2+}$ transients (*Figure 5B*). PKC$^-$ networks realized much higher PFRs and had somewhat smaller exponents than PKC$^N$ (PKC$^N$0.12 ± 0.02, PKC$^-$0.11 ± 0.01, p=3.2*10$^{-18}$; *Figure 5C,D, E*).

In all network types, PFR increased steeply in early development and later declined concurrently with SBE strength. Throughout development, however, PFRs were highest in homogeneous networks and lowest in clustered networks (*Figure 5F*, *Table 2*). Networks with low AFR thus had high PFR.

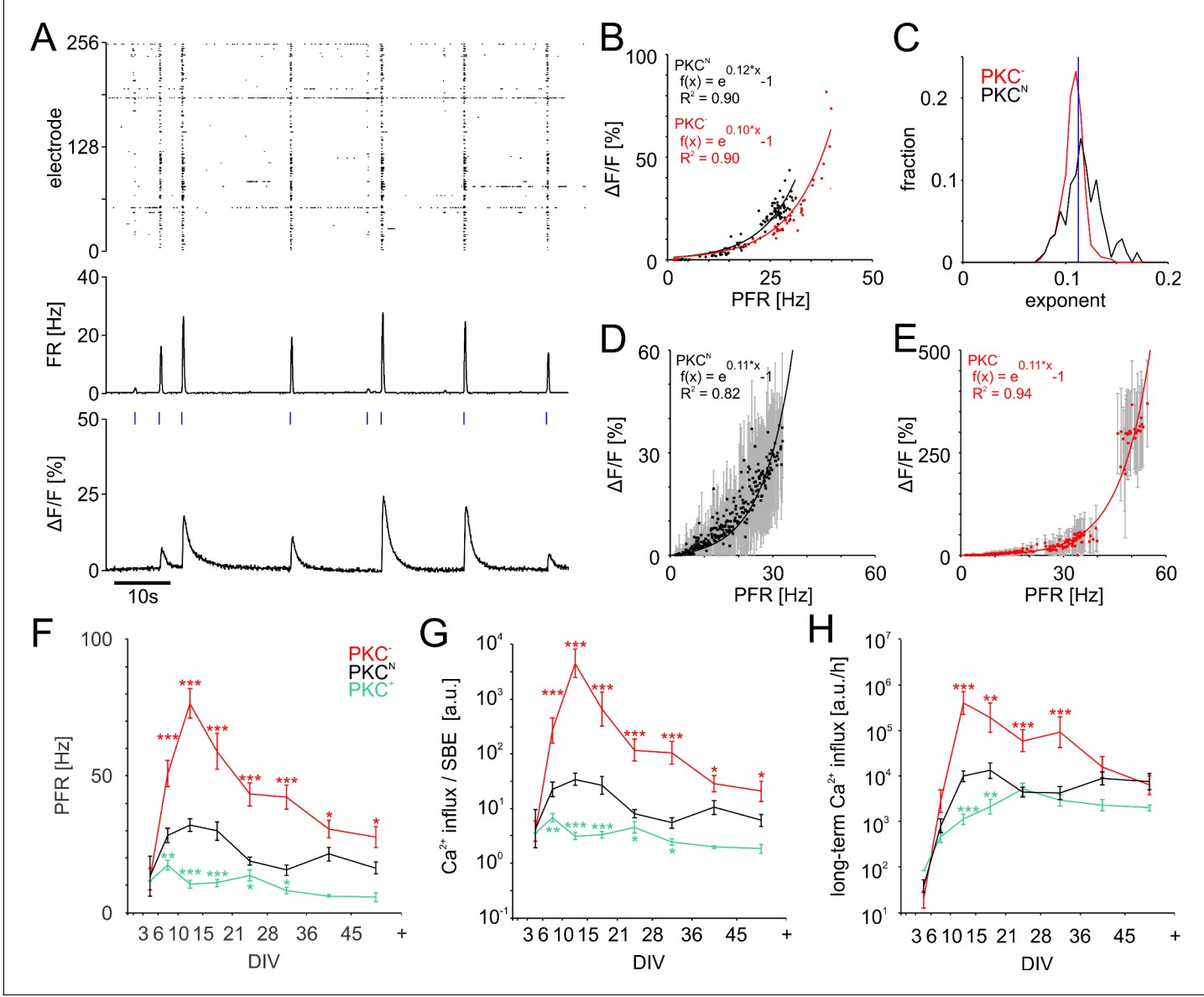

**Figure 5.** PFR-dependent Ca$^{2+}$ gain. (A) Spiking raster and firing rate averaged across all electrodes within 40 ms bins (synchronized to the frame times of the Ca$^{2+}$ measurement) and Ca$^{2+}$ signal for one neuronal soma at 19 DIV in a PKC$^N$ network. *Blue ticks*: SBE onsets. (B) The amplitude of Ca$^{2+}$ transients (shown for the PKC$^N$ neuron in A and a PKC$^-$ neuron at 20 DIV) scaled exponentially (solid line) with PFR. (C) Exponents had a narrow distribution and were slightly higher in PKC$^N$ (p=3.2*10$^{-18}$) than in PKC$^-$ conditions (PKC$^N$: 179 neurons, 5 networks at 19 DIV, 714 SBEs total, mean and standard deviation of exponent = 0.12 ± 0.02; PKC$^-$: 622 neurons, 4 networks at 20 DIV, 248 SBEs total, exponent = 0.11 ± 0.01). *Blue*: average exponent (0.11 ± 0.01) for the entire data set. (D) Ca$^{2+}$ amplitudes scaled exponentially with PFRs across many neurons in these PKC$^N$ and PKC$^-$ networks (E). The data represent median (*data points*) and standard deviation (*error bars*) of Ca$^{2+}$ amplitudes, averaged across neurons for a given PFR range (bin size 0.1 Hz). (F) PFR assessed during SBEs were higher in homogeneous networks and lower in clustered networks. PFR decreased after week 3, putatively with the maturation of inhibition. (G) Prediction of the development of average Ca$^{2+}$ influx per SBE estimated as $e^{0.11*PFR} - 1$ (*Figure 5D*). (H) Average Ca$^{2+}$ influx per minute, estimated from all SBEs in 1 hr recording sessions, suggests that long-term average Ca$^{2+}$ influx in different PKC conditions converged at network maturation. Data in G and H are presented as mean ± SEM. Asterisks indicate p-values ≤0.05 (*), ≤0.01 (**) and ≤0.001 (***) tested against PKC$^N$.

DOI: https://doi.org/10.7554/eLife.47996.021

The following source data is available for figure 5:

**Source data 1.** Source data and Matlab script for *Figure 5B–E,F*.

DOI: https://doi.org/10.7554/eLife.47996.022

Knowing the relationship between PFR and $Ca^{2+}$ influx allowed us to estimate $Ca^{2+}$ levels during development based on MEA recordings. We approximated the development of the average $Ca^{2+}$ influx per SBE (*Figure 5G*) from their respective PFRs and the exponential $Ca^{2+}$ gain function with the average exponent of 0.11. Because higher PFRs, $Ca^{2+}$ influx per SBE was highest in the more homogeneous $PKC^-$ networks and lowest in clustered $PKC^+$ networks. Yet, in combination with the systematic increase of SBE rate with clustering, long-term $Ca^{2+}$ influx converged during late development for different PKC conditions, network architectures and AFR (*Figure 5H*).

## Differences in PFR reflect variations of network recruitment during SBEs

The predominately short-range connectivity observed in clustered $PKC^+$ networks could impair network-wide recruitment (*Okujeni et al., 2017*) and synchronization of activity. This would decorrelate inputs, explaining lower PFR and weaker membrane depolarization during SBEs. To test this, we determined network synchrony as the average spike correlations between all electrode pairs (*Figure 6A*). Consistent with the rapid buildup of connectivity, network synchrony increased steeply between 3–15 DIV and reached stable levels already between 15–21 DIV, even though activity levels, connectivity and inhibition continued to develop. In line with connectivity estimates, synchronization was highest in $PKC^-$ networks ($0.53 \pm 0.04$, $p=4.6*10^{-5}$ compared to $PKC^N$), intermediate in $PKC^N$ networks ($0.35 \pm 0.02$) and lowest in $PKC^+$ networks ($0.16 \pm 0.03$, $p=1.7*10^{-5}$ compared to $PKC^N$), that is network synchrony indeed decreased with the degree of clustering.

## Maturation of inhibition is comparable across PKC conditions

Neural development involves a transition from excitatory to inhibitory GABAergic transmission by upregulation of the $Cl^-$-transporter KCC2. This maturation of inhibition considerably influences

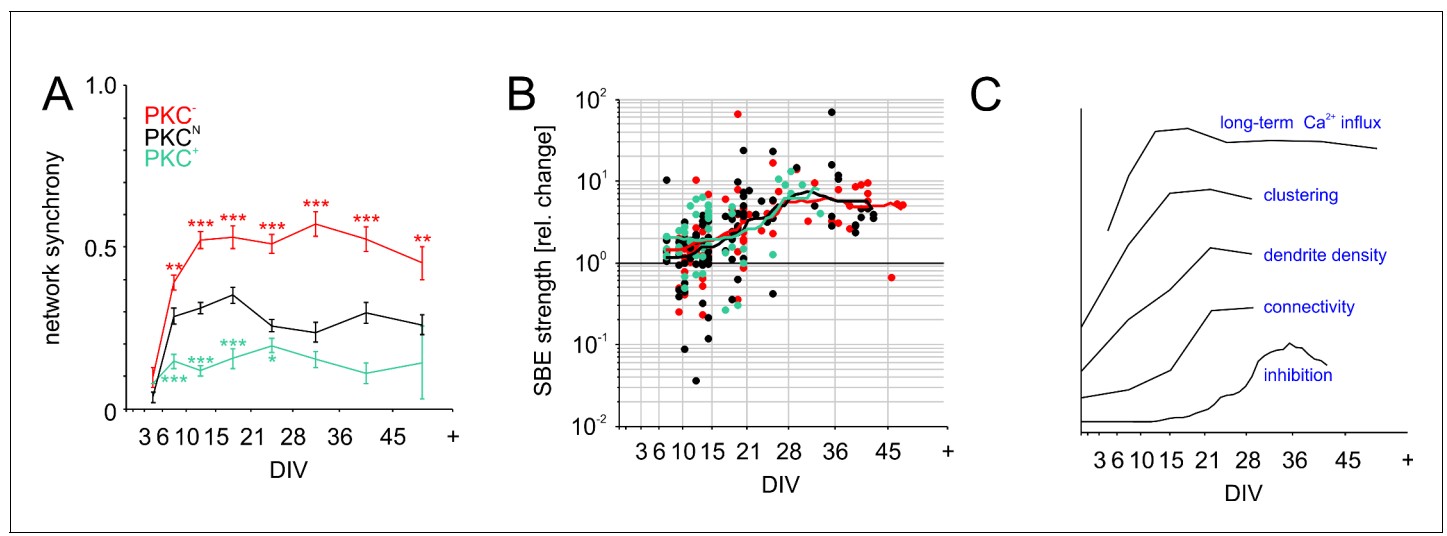

**Figure 6.** Functional aspects of network maturation. (**A**) Network synchrony (average spike train correlation for all electrode pairs determined with 30 ms time bins, mean ± SEM) stabilized early in development in all PKC conditions but was significantly higher in homogeneous networks and significantly lower in clustered networks. Asterisks indicate p-values ≤0.05 (*), ≤0.01 (**) and ≤0.001 (***) tested against $PKC^N$. (**B**) Inhibition was probed by acute blockade of GABA-A receptors with PTX. In early networks, PTX had highly variable impact on SBE strength (<14 DIV) but significantly increased it after 21 DIV in all network types. The maturation of inhibition was comparable across PKC conditions. (**C**) Illustration of the time course of key differentiation processes shown in detail in *Figures 3* and *5* in relation to the development of long-term average $Ca^{2+}$ influx. Morphological parameters that showed a similar time course across PKC conditions were normalized to final levels to visualize the relative change during development. The maturation of inhibition is shown as average relative change in SBE strength upon PTX appliction. Y-axis scaling is linear for all graphs. An offset was added for visualization.

DOI: https://doi.org/10.7554/eLife.47996.023

The following source data is available for figure 6:

**Source data 1.** Source data and Matlab script for *Figure 6A,B*.
DOI: https://doi.org/10.7554/eLife.47996.024

activity levels and dynamics, while being activity-dependent itself (*Fiumelli and Woodin, 2007*). Inhibition crucially affects network activity and thus interacts with $Ca^{2+}$ influx and neuronal morphogenesis. Furthermore, since PKC promotes membrane incorporation of KCC2, reducing its activity could delay the maturation of inhibition and thus indirectly influence activity-dependent network development. To test if PKC manipulation altered the maturation of GABAergic inhibition on the network level, we blocked GABA-A receptor-dependent transmission at different developmental stages (10 µM PTX; *Figure 6B*) and recorded the resulting change of network activity with MEAs (number of recorded networks between 8–48 DIV; $PKC^N$: N = 88; $PKC^-$: N = 85; $PKC^+$: N = 48). Acute application of PTX had variable impact on SBE strength up to 14 DIV ($PKC^N$: +76 ± 31%, p=$4.0*10^{-1}$, N = 33; $PKC^-$: +62 ± 28%, p=$3.6*10^{-3}$, N = 33; $PKC^+$: +120 ± 45%, p=$2.0*10^{-3}$, N = 24; mean ± SEM, paired Student's t-test) but significantly amplified bursting after 21 DIV ($PKC^N$: +785 ± 159%, p=$1.4*10^{-5}$, N = 31; $PKC^-$: +455 ± 103%, p=$3.8*10^{-10}$, N = 29; $PKC^+$: +524 ± 188%, p=$3.7*10^{-2}$, N = 11). The developmental time course of the average PTX impact was comparable across PKC conditions, indicating a comparable maturation of GABAergic inhibition. We therefore concluded that the observed alterations in network dynamics were the result of differences in network architecture rather than the result of differences in inhibition levels.

## Discussion

Neuronal network architecture is not based on a genetic blueprint alone but is shaped by predefined rules of activity-dependent self-organization (*Spitzer, 2006*). Herein, neuronal migration (*Komuro and Kumada, 2005*; *Zheng and Poo, 2007*) and neurite outgrowth (*Kater et al., 1988*) are regulated by activity-related changes of $[Ca^{2+}]_i$. Indeed, cell motility and growth is optimal within a narrow $[Ca^{2+}]$ range and diminished otherwise, which led to the hypothesis that network connectivity and activity evolve under homeostatic control with the $[Ca^{2+}]_i$ as set-point parameter (*Kater and Mills, 1991*). However, basal cytosolic $[Ca^{2+}]$ is very low due to efficient $Ca^{2+}$-buffering and extrusion (*Kater and Mills, 1991*; *Zündorf and Reiser, 2011*) and remains relatively constant during development (*Maravall et al., 2000*). Free $Ca^{2+}$ for the regulation of growth is thus essentially determined by transient $[Ca^{2+}]_i$ elevations induced by synaptic input and spike activity. Accordingly, the developmentally attained spike rate was proposed to reflect the $Ca^{2+}$ set-point of growth (*van Ooyen et al., 1995*).

The overall capacity for neurite growth ultimately relies on gene expression for cytoskeletal building blocks, which crucially depends on nuclear $[Ca^{2+}]$ (*Berridge et al., 2000*). Somatic membrane depolarization increases $Ca^{2+}$ influx close to the nucleus (*Greer and Greenberg, 2008*). In this context, intracellular stores like the endoplasmic reticulum can accumulate $Ca^{2+}$ over longer periods of time and then considerably amplify $Ca^{2+}$ signals by additional $Ca^{2+}$-triggered release (*Berridge et al., 2000*; *Pivovarova et al., 2002*). This effectively acts as a low-pass filter and amplifier for $Ca^{2+}$-signaling to the nucleus – modulating the expression of cytoskeletal proteins. Supporting this link, neurite tree morphology and size in different neuron types appear to depend on the expression of specific $Ca^{2+}$-binding proteins that determine nuclear $Ca^{2+}$ buffering capacity (*Mauceri et al., 2015*). In contrast to nuclear $Ca^{2+}$ levels, local $Ca^{2+}$ transients in neurites direct migration and growth towards target neurons (*Guan et al., 2007*; *Henley and Poo, 2004*; *Hutchins and Kalil, 2008*), which promotes neurite overlap and synaptic connectivity (*Shepherd et al., 2005*; *Stepanyants et al., 2002*). Though local $Ca^{2+}$ influx activating PKC modulates cytoskeletal turnover involved in guided outgrowth and migration (*Fogh et al., 2014*; *Kabir et al., 2001*; *Larsson, 2006*), PKC may not be essential for constitutive neurite outgrowth (*Flynn, 2013*; *Letourneau et al., 1987*). We therefore speculate that local $Ca^{2+}$ transients and PKC activity regulate cytoskeletal motility to direct growth processes, whereas long-term accumulation of $Ca^{2+}$ in intracellular stores modulates signaling to the nucleus, transcription levels and thus the overall availability of cytoskeletal building blocks. This predicts a cessation of growth at a target long-term average $Ca^{2+}$ influx that is independent from PKC activity.

### Migration contributes to homeostatic network development

Extending on growth models for homeostatic network formation based on activity-dependent neurite outgrowth, neuronal migration could likewise contribute to the regulation of connectivity and activity in developing networks. *Eglen et al. (2000)* already added migration implemented as

repulsion between neurons to the neurite growth model by *van Ooyen et al. (1995)* to generate regular neuronal arrangements as observed in dense retinal cell mosaics. We showed that activity-dependent attraction between migrating neurons leads to different degrees of modularity by the interaction of clustering with homeostatic regulation of neurite growth. While it is plausible to assume that neurons with small neurite fields and little connectivity may move, this seems less realistic once they are enmeshed in the network. In line with this, cell migration relies on localized $Ca^{2+}$ transients in leading neurites and the resulting $Ca^{2+}$ gradients across the cell (*Guan et al., 2007*) but ceases with increasing neuronal activity and frequency of $Ca^{2+}$ transients (*Bando et al., 2016*). We approximated this in the model by allowing attraction while input was below the set-point but omitted repulsion with input above the set-point. In consequence, cell migration ended during the rapid increase of activity during development, similar to peaks in PFR and $Ca^{2+}$ influx, and cessation of clustering around 10–15 DIV (*Figure 3B*, *Figure 5F,H*). Moreover, with rapid transitions to high network activity once neurite fields in the network overlapped sufficiently, the model showed a transient overshoot of connectivity. A more gradual build-up of activity diminished the average overshoot and pruning when the slope of the sigmoid mapping input to firing rate was reduced (*Figure 2—figure supplement 2*), in agreement with reports of varying degrees of growth overshoot or even saturating growth during development in vitro (*Ito et al., 2013*; *Kondo et al., 2017*; *van Pelt et al., 2004*). Neurons that connected to the network early in development, however, still showed an overshoot of connectivity, in agreement with *Kossio et al. (2018)*.

## Average $Ca^{2+}$ influx converges for different network architectures

Homeostatic regulation of growth processes by $Ca^{2+}$ was proposed to guide network development towards target firing rates (*van Ooyen et al., 1995*), which implies a quasi-linear relationship between $Ca^{2+}$ influx and AFR. In our model, connectivity, input activity and firing rates eventually converged to the same levels for different migration conditions and network architectures. In apparent conflict with the simulation, we found that different network architectures stabilized in vitro after about 3 weeks but with different AFR. Consistent with theoretical studies predicting that network modularity promotes spontaneous activity (*Kaiser and Hilgetag, 2010*; *Klinshov et al., 2014*; *Mazzucato et al., 2015*), SBE rates and AFRs increased with the level of clustering. Clustering, however, reduced network synchronization, lowered PFRs and weakened depolarization during SBEs. This strongly affected $Ca^{2+}$-transients: $Ca^{2+}$ peak amplitude increased exponentially with PFR during SBEs, in agreement with reports of $Ca^{2+}$ currents through voltage-gated $Ca^{2+}$ channels increasing exponentially with depolarization (*Mayer et al., 1987*; *Mazzanti et al., 1992*). Because of the opposite modulation of SBE rates and PFRs with clustering, however, the estimated long-term $Ca^{2+}$-gain converged for different network architectures during development, despite different AFR. The low spike rates during inter-burst intervals had negligible influence on $Ca^{2+}$ influx.

To account for the supra-linear increase of $Ca^{2+}$ with PFR we would need to use spiking neurons in our model. In addition, $Ca^{2+}$ influx would need to depend on the membrane potential, instead of on the average spike rate of a neuron as in extensions of the growth model with spiking dynamics (*Abbott and Rohrkemper, 2007*; *Kossio et al., 2018*). To accelerate the simulation of several weeks of network development, these studies initially increased the neurite growth rate and thus effectively decreased the temporal resolution until the networks approached the equilibrium state. The mesoscale structures forming in our networks, however, crucially depended on the continuous feedback between migration and neurite growth and activity. Low temporal resolution in the simulation would amount to a large decrease of the feedback speed, which leads to a random walk of neurons and more homogeneous network structures without clustering.

## Interaction between growth and migration shapes network modularity

Increasing the rate of activity-dependent migration in the model promoted clustering, decreased neurite fields and accelerated the development of spontaneous activity by more rapidly increasing neurite overlap and connectivity. This resulted in network architectures covering a continuous gradient from homogeneous via partially clustered with scattered neurons to fully clustered networks with corresponding degrees of modularity. This was remarkably similar to the development in vitro, where PKC activity promoted clustering and SBE rates, and decreased neurite density. The model

suggests that different network architectures can arise spontaneously based on simple rules regulating connectivity to achieve a target level of $[Ca^{2+}]_i$.

Among the grand average developmental time courses of the most relevant aspects across all conditions, long-term $Ca^{2+}$ influx was the first property to peak while the impact of inhibition on network activity only started to increase when $Ca^{2+}$ influx stabilized (*Figure 6C*).

## Growth and migration shape the framework for synaptic connectivity

In our networks, synapse densities scaled approximately quadratically with the average dendrite size and thus negatively with the degree of clustering. This could be explained by the co-modulation of axonal and dendritic densities in the same direction (*Okujeni et al., 2017*), which multiplicatively increases the number of axo-dendritic contact sites, rather than their modulation in opposite directions as used in *Tetzlaff et al. (2010)*. Such potential synapses realize into functional synapses with approximately constant probability in vivo (*Stepanyants et al., 2002*). The consistent relation of synapse density and dendrite size across developmental stages and PKC conditions (*Figure 3G*) suggests that PKC manipulation did not critically impair synaptogenesis. Our estimates of maximum connectivity suggest a saturation of connectivity towards 10% in clustered and 20% in homogeneous networks, in the range of values reported for cultured (*Marom and Shahaf, 2002*) and native cortical networks (*Feldmeyer, 2012*).

The mesoscale network architecture formed early thus appears to determine the probabilistic framework for connectivity. PKC activity additionally influences synaptic plasticity, yet without general directionality towards LTP or LTD (*Chung et al., 2000*; *Ferreira et al., 2011*; *Lan et al., 2001*; *Boehm et al., 2006*; ; *Scott et al., 2007*). Our model indirectly accommodates this influence. For example, synaptic depression, corresponding to reducing the synaptic weight factor *s*, would extend the outgrowth phase to increase connectivity and input necessary to reach the target level of $[Ca^{2+}]_i$. Conceptually, this would be the inverse of the homeostatic scaling of synaptic weights with the level of connectivity (*Barral and D Reyes, 2016*; *Okujeni et al., 2017*; *Wilson et al., 2007*). This contribution of synaptic plasticity to the activity-dependent fine-tuning of connectivity likely gains importance with increasing developmental age and structural complexity of a network.

## Conclusion

Based on our findings, we propose that interactions between neurite growth and neuronal migration affect the balance between local and global connectivity, thereby shaping network modularity. Cell migration defects were also proposed as a pathogenic mechanism involved in several neurological conditions associated with altered size and spacing of mini-columns in the cortex, aberrant neurite growth and hyper- or hypo-connectivity (*Catts et al., 2013*; *Courchesne and Pierce, 2005*; *Di Rosa et al., 2009*; *Donovan and Basson, 2017*; *Fan et al., 2013*; *McKavanagh et al., 2015*), suggesting that the mesoscale network organization could be a critical factor. The associated degree of modularity thus appears to have crucial impact on activity generation, propagation and perpetuation, neural synchronization as well as network function and dysfunction.

# Materials and methods

**Key resources table**

| Reagent type (species) or resource | Designation | Source or reference | Identifiers | Additional information |
|---|---|---|---|---|
| Strain, strain background (Rattus norvegicus domestica) | wildtype wistar rat pups | CEMT, University, Freiburg | | |
| Genetic reagent | AAV9.CAG.GCaMP6s.WPRE.SV40 | Penn Vector Core, University of Pennsylvania | V3296TI-R | titer 1e11 |
| Antibody | anti-MAP2 (chicken polyclonal) | Abcam, Cambridge, UK | ab92434 RRID: AB_2138147 | 1:500 |

*Continued on next page*

*Continued*

| Reagent type (species) or resource | Designation | Source or reference | Identifiers | Additional information |
|---|---|---|---|---|
| Antibody | anti-NeuN (rabbit polyclonal) | Abcam, Cambridge, UK | ab128886 RRID:AB_2744676 | 1:500 |
| Antibody | anti-Neurofilament (mouse monoclonal) | Abcam, Cambridge, UK | ab24571 RRID:AB_448148 | 1:10 |
| Antibody | anti-Synapsin (mouse monoclonal) | Synaptic Systems GmbH, Germany | 106001 RRID:AB_887805 | 1:200 |
| Antibody | anti-chicken-Cy2 (goat polyclonal) | Abcam, Cambridge, UK | ab6960 RRID:AB_955003 | 1:200 |
| Antibody | anti-rabbit-Cy3 (goat polyclonal) | Abcam, Cambridge, UK | ab6939 RRID:AB_955021 | 1:200 |
| Antibody | anti-mouse-Cy5 (goat polyclonal) | Abcam, Cambridge, UK | ab6563 RRID:AB_955068 | 1:200 |
| Chemical compound, drug | 4,6-diamidino-2-phenyindole, diclactate (DAPI) | Sigma-Aldrich, Germany | D9562 | 1:5000 |
| Chemical compound, drug | Gödecke6976 | Tocris Bioscience, Bristol, UK | 2253 | 1 µM |
| Chemical compound, drug | Phorbol-12-Myristate-13-Acetate (PMA) | Sigma-Aldrich, Munich, Germany | P1585 | 1 µM |
| Chemical compound, drug | Picrotoxin | Tocris Bioscience, Bristol, UK | 1128 | 10 µM |
| Chemical compound, drug | DMSO | Sigma-Aldrich, Munich, Germany | D8418 | 0.1% |
| Chemical compound, drug | DNase (type IV) | Sigma-Aldrich, Munich, Germany | D5025 | 50 g/ml |
| Chemical compound, drug | minimal essential medium | Invitrogen, Karlsruhe, Germany | 21090055 | |
| Chemical compound, drug | horse serum (heat-inactivated) | Invitrogen, Karlsruhe, Germany | 26050088 | 20% |
| Chemical compound, drug | phosphate buffered saline (PBS) | Invitrogen, Karlsruhe, Germany | 21600010 | |
| Chemical compound, drug | glucose | Sigma-Aldrich, Munich, Germany | G7528 | 20 mM |
| Chemical compound, drug | L-glutamine | Invitrogen, Karlsruhe, Germany | 25030024 | 0.5 mM |
| Chemical compound, drug | gentamycin | Invitrogen, Karlsruhe, Germany | 15750060 | 20 µg/ml |
| Chemical compound, drug | potassium D-gluconate | Sigma-Aldrich, Munich, Germany | G4500 | 125 mM |
| Chemical compound, drug | EGTA | Carl Roth, Karlsruhe, Germany | 3054 | 5 mM |
| Chemical compound, drug | KCl | Sigma-Aldrich, Munich, Germany | P4504 | 20 mM |
| Chemical compound, drug | $Na_2$-ATP | Carl Roth, Karlsruhe, Germany | K054 | 2 mM |
| Chemical compound, drug | Hepes | Carl Roth, Karlsruhe, Germany | 9105 | 10 mM |
| Chemical compound, drug | $CaCl_2$ | Sigma-Aldrich, Munich, Germany | C3881 | 0.5 mM |
| Chemical compound, drug | KOH | Sigma-Aldrich, Munich, Germany | P4504 | |

*Continued on next page*

*Continued*

| Reagent type (species) or resource | Designation | Source or reference | Identifiers | Additional information |
|---|---|---|---|---|
| Chemical compound, drug | MgCl$_2$ | Sigma-Aldrich, Munich, Germany | MO250 | 2 mM |
| Software | MC Rack software | Multi Channel Systems, Germany | versions 3.3–4.5 RRID:SCR_014955 | |
| Software | Spike2 software | Cambridge Electronics Design Ltd., Cambridge, UK. | RRID:SCR_000903 | |
| Software | Zen | Carl Zeiss, Jena, Germany | RRID:SCR_013672 | |
| Software | MEA-Tools | *Egert et al., 2002* (PMID 12084562) | version 2.8 | |
| Software | FIND toolbox | *Meier et al., 2008* (PMID 18692360) | | |
| Software | ImageJ | *Schneider et al., 2012* (PMID 22930834) | RRID:SCR_003070 | |
| Software | Matlab | Mathworks, Natick, MA, USA | versions 2014a – 2017a | |

## Network growth model

We adopted and modified the model of activity-dependent network growth introduced by *van Ooyen et al. (1995)*. All simulations were carried out with Matlab (version 2017a, Mathworks, Natick, MA, USA; code available at doi 10.5281/zenodo.3459678).

Networks were initialized by randomly seeding 500 neurons onto a torus surface of 1 mm$^2$ to avoid boundary effects. Newly introduced neurons conflicting with the minimal neuron distance of 12 µm, approximately the size of cell bodies, were discarded and the procedure continued until the required neuron density was obtained.

Neurite fields were modeled as circular fields, centered at cell bodies and were initiated with a radius of 12 µm. Connectivity between neurons $W$ was nonsymmetrical and defined as the area $A$ of neurite field overlap normalized by the area of the presynaptic neuron, which reflected the probability that dendrites of neuron $i$ overlapped with the axons of presynaptic neuron $k$.

$$W_{ki} = s\frac{A_i \cap A_k}{A_k}$$

The gain $s = 0.1$ was chosen such that it produced networks with an intermediate degree of neurite field overlap (for $s = 1$, neurons would only connect to one or a few other neurons). Instead of simulating network growth with dimensionless equations (*van Ooyen et al., 1995*), we adjusted the time steps such that we could compare the dynamics to realistic developmental timescales. We estimated the loop-time across which activity is integrated based on the time constants for the accumulation of Ca$^{2+}$ in intracellular stores to be in the order of minutes (*Pivovarova et al., 2002*) and therefore set the temporal resolution of the simulation to 1 min.

Since inhibition is not explicitly relevant to the questions addressed here, we adapted the model for excitatory networks only. Long-term integration of activity in neurons was described by their state variable $x_i$ (ranging between 0 and 1), which increased with input from presynaptic neurons contributing with their firing rate $f(x_k)$ times the synaptic strength $W_{ki}$:

$$\frac{dx_i}{dt} = -\frac{x_i}{\tau} + (1-x_i)\sum_{k}^{N} W_{ki}f(x_k)$$

where $dt = \tau = 1$ min was the time resolution of the simulation, corresponding to the time constant of long-term integration of activity. A sigmoidal transfer function for the depolarization state $x_i$ determined the firing rate $f(x)$.

$$f(x_i) = \frac{1}{1 + e^{(\theta - x_i)/a}}$$

where $\theta$ = 0.5 reflected the firing threshold and $a$ = 0.12 determined the steepness of the function that crucially impacted on the developmental overshoot of connectivity and subsequent pruning of neurites. We chose a slightly shallower function than the original model by Van Ooyen ($a$ = 0.1) to accommodate the degree of overshoot and pruning for cultured networks in recent reports (*Ito et al., 2013*; *Kondo et al., 2017*; *van Pelt et al., 2004*).

As in the original model by Van Ooyen, neurons were modeled to grow neurites and thereby increase input activity and firing rate to reach a target $[Ca^{2+}]_i$. If this $Ca^{2+}$ level was surpassed, neurites were pruned, in turn. These bidirectional changes in the radius $R$ of circular neurite fields, were determined by a sigmoidal function of the firing rate of a neuron multiplied with a fixed growth rate $\rho_{growth}$.

$$\frac{dR_i}{dt} = 1 - \frac{2}{1 + e^{\frac{(\varepsilon - f(x_i))}{\beta}}} \rho_{growth}$$

where $\varepsilon$ = 0.6 defined the target level for activity or $[Ca^{2+}]_i$, $\beta$ = 0.1 determined the steepness of the sigmoidal function and $\rho_{growth}$ was the constant factor for the growth rate of neurite fields. We assumed that connectivity is mainly determined by the density of neurites rather than their maximal length. Given the homogeneous density of the neurite field used in the model, however, its radial expansion must be considerably slower than the average elongation rates of individual dendrites, which were reported to be 12 µm/day for isolated neurons in the first week in vitro (*Mattson and Kater, 1988*). We therefore set $\rho_{growth}$ = 4 µm per day.

In our model, neurons additionally migrated in the direction of presynaptic inputs, thus mimicking the guidance of migration by leading processes (*Flynn, 2013*; *Guan et al., 2007*) and consistent with the positive correlation between the rate of soma translocation and the amplitude and frequency of $Ca^{2+}$ transients (*Komuro and Kumada, 2005*; *Zheng and Poo, 2007*). We assumed synaptic activity in leading neurites as an important source of input, however, did not preclude contact-mediated $Ca^{2+}$ signaling (*Sheng et al., 2013*), which may contribute in regulating migration early in development when activity levels are low. Changes in the spatial position of neuronal cell bodies S were caused by migration impulses that depended on $[Ca^{2+}]_i$ and, thus, on the firing rate $f$ and a variable factor for the maximal migration rate $\rho_{migration}$.

$$\frac{dS_i}{dt} = e^{f(x_i)} \rho_{migration}$$

where $\rho_{migration}$ ranged 0-300 µm/day and µ = -15 determined how strong migration impulses were diminished as neurons reached their target $Ca^{2+}$ level. We chose µ to result in a negligible migration impulse at the target $[Ca^{2+}]_i$. This mimicked a realistic migration process in which neurons are guided by local $Ca^{2+}$ transients in leading neurites and the resulting $Ca^{2+}$ gradients across the cell (*Guan et al., 2007*), but at the same time cease migrating when spiking-based $Ca^{2+}$ transients start to dominate (*Bando et al., 2016*). The migration speed of postnatal neurons in vitro indeed decays approximately exponentially during development from 0.7 µm/min (1008 µm/day) at 0 DIV to ~0.05 µm/min (72 µm/day) at 12 DIV on Matrigel-coated substrates and with slower initial migration speeds of 0.1 µm/min (144 µm/day) on PEI coated substrates (*Sun et al., 2011*), as used in this study. In the model we varied migration rates within this range.

The direction of movement was determined involving a directed movement component and a random movement component to match erratic movements observed in time lapse videos. Movement direction of the directed component was determined by the vector sum $v_{dir}$ of direction vectors $v_{ik}$ that pointed to presynaptic neurons and were weighted by their input.

$$v_{dir} = - \sum_{k}^{N} W_{ki} f(x_k) v_{ik}$$

To obtain the final direction vector $V$, directed and the random component (updated every 10

min) were weighted ($p$ = 0.9) and summed. The random directional component was necessary to mimic the erratic movement patterns observed in in vitro time lapse studies.

$$V = \frac{v_{dir}}{\|v_{dir}\|}(1-p) + \frac{v_{rand}}{\|v_{rand}\|}p$$

New neuronal cell body positions $P$ were determined by multiplying the normalized final direction vector with the migration impulse.

$$P(x,y)_{new} = P(x,y)_{old} + \frac{v}{\|v\|} \cdot \frac{dS_i}{dt}$$

In addition, neurons were set to jitter randomly around their current position by maximally their cell body radius to allow neurons to pass each other in the 2D simulation, which prevented unrealistic chains of neurons. This positional jitter decreased according to the exponential decay function modulating migration in dependence of $[Ca^{2+}]_i$ such that neurons stopped moving when reaching the target value. It was reset after each time step. Movements violating the minimal possible inter-soma distance (12 μm) were discarded.

To assess the modularity of a network, we calculated the size of the largest subnetwork (the giant component) remaining after removing defined fractions of randomly selected neurons from the network as its fraction in the remaining total population. For each network, the results were averaged across 1000 repetitions of the procedure. We quantified the degree of modularity Q in the final networks based on the connectivity matrix using the Louvain method (*Blondel et al., 2008*) implemented for MATLAB by Mika Rubinov with gamma = 1 (*Rubinov and Sporns, 2010*). Q increases towards one with increasing modularity. Random networks yield Q = 0.

## Cell culture techniques

Primary cortical cell cultures were prepared on different MEAs (Multi Channel Systems, Reutlingen, Germany (MCS); electrode grid layout/pitch distance (μm): 8 × 8/200; 6 × 10/500; 16 × 16/200) and standard coverslips (12 mm diameter, Carl Roth, Karlsruhe, Germany). All substrates were coated with polyethylene-imine (150 μl 0.2% aqueous solution; Sigma-Aldrich, Munich, Germany) for cell adhesion. Cell cultures were prepared following (*Shahaf and Marom, 2001*). Cortical tissue was prepared from brains of neonatal Wistar rat pups of either sex, minced with a scalpel and transferred into phosphate buffered saline (Invitrogen, Karlsruhe, Germany). Tissue pieces were incubated with trypsin (isozyme mixture, 0.05%, 37°C, 15 min; Invitrogen). proteolysis was stopped with horse serum (20%; Invitrogen). DNase (type IV, 50 μg/ml; Sigma-Aldrich) was added to eliminate cell trapping in DNA strands if needed. Cells were dissociated by trituration with a serological pipette, centrifuged (5 min, 617 g) and resuspended in growth medium (Minimal Essential Medium supplemented with 5% heat-inactivated horse serum, 0.5–1 mM L-glutamine and 20 μg/ml gentamycin (all from Invitrogen), 20 mM glucose (Sigma); 1 ml/pup). Cells were counted with an automated cell counter (CASY, Schärfe Systems, Reutlingen, Germany) and seeded with ~300.000 cells per network (~1 cm$^2$). Sparse cultures for morphological analysis were seeded with ~37.500 cells per network. Networks developed in 1 ml growth medium in a humidified incubator (5% $CO_2$. 37°C). Animal handling and tissue preparation were done in accordance with the guidelines for animal research at the University of Freiburg and approved by the Regierungspräsidium Freiburg (permits X-12/08D, X-16/07A, X-15/01H, X-18/04K).

## PKC modulation and disinhibition

PKC inhibitor Gödecke6976 (Gö6976, 1 μM; Tocris Bioscience, Bristol, UK) and PKC agonist Phorbol-12-Myristate-13-Acetate (PMA, 1 μM; Sigma-Aldrich) were dissolved in dimethyl sulfoxide (DMSO, Sigma-Aldrich) and added to the culture medium directly after cell preparation. The maximal concentration of DMSO in the growth medium was 0.1%. GABAergic transmission was probed by acute application of the non-competitive GABA-A receptor antagonist Picrotoxin (PTX; 10 μM; Tocris Bioscience) during electrophysiological recordings. Recordings of spontaneous activity were started 10 min after application of PTX for 1 hr at different DIV. Changes of spike activity were calculated as mean burst strength across 1 hr with PTX vs. 1 hr baseline recording before application. Networks exposed to PTX were discarded.

## Morphological analyses

The development of neuronal clustering, dendrite outgrowth and synapse densities was analyzed in sparse networks of ~100 neurons/mm$^2$ that were more accessible for quantitative morphological analysis. Clustering of neuronal cell bodies was analyzed based on immunocytochemical staining of neuronal nuclei (NeuN; Rabbit-anti-NeuN, 1:500; Abcam, Cambridge, UK, RRID:AB_2744676) and of all cellular nuclei (DAPI; Sigma-Aldrich). Neuronal nuclei were detected based on NeuN and DAPI colocalization and evaluated for their degree of clustering using a modified Clark-Evans clustering index (CI) that accounts for cell body diameter as minimal possible inter-neuron distance (*Clark and Evans, 1954*; *Galli-Resta et al., 1999*; *Okujeni et al., 2017*). CI was calculated as the ratio between the average nearest neighbor distance in a network and the expected average nearest neighbor distance for random networks. Note that the degree of clustering increases with decreasing CIs below 1. CIs above one indicate grid-like cell body arrangements. Dendrite morphology was examined by immunocytochemical staining of microtubule-associated protein 2 (MAP2, Chicken-anti-MAP2; 1:500; Abcam, RRID:AB_2138147). To quantify the total length of dendrites, MAP2 images taken at 20-fold magnification (0.323 μm/pixel) were processed by median filtering (3 × 3 kernel), background subtraction (lowest value in 7 × 7 pixel field), contrast adjustment (saturation at highest and lowest 10%), thresholding and skeletonization of the resulting binary image, similarly to *Pani et al. (2014)*. Synapses were detected based on an immunohistological staining of the presynaptic protein synapsin (Mouse-anti-Synapsin; 1:200; Synaptic Systems GmbH, Göttingen, Germany, RRID:AB_887805). Synaptic punctae were then determined by local maximum detection in high-pass filtered and contrast-enhanced images. We analyzed two networks per condition and age taken from images covering approximately 3.5 mm$^2$. In each image, we typically analyzed 10–20 regions of interest with varying size (could overlap) and including dense and sparse network regions. The following measures were determined as the slope of the linear regression through data pairs from all regions of interest: *Dendrite size*, total length of dendrite stretches relative to the number of neurons; *Synapse density*: average number of synapses relative to the number of neurons; *Dendritic occupancy*: average number of synapses relative to the total length of dendrite stretches; Neuron density, average number of neurons per area; *Maximum connectivity*, ratio between the number of synapses per neuron and the total number of neurons in the network (extrapolated for the entire network area of ~1.1 cm$^2$ given the image neuron density). All morphometric analyses were done with Matlab (versions 2014a – 2017a). Results are presented as mean ± standard error of the mean (SEM) and significance was assessed with a two-tailed independent Student's t-test. Network architectures of dense networks (600–800 neurons/mm$^2$) were characterized qualitatively at 22 DIV with antibodies against MAP2 and phosphorylated neurofilament 200 kD (Rabbit-anti-neurofilament; 1:10; Abcam, RRID:AB_448148) to visualize dendritic and axonal compartments, respectively.

## Extracellular recording and analyses

MEA recordings (MEA1060-BC and USB-MEA256-Systems; MCS, 25 kHz sampling frequency, 12 bit AD-conversion; MCRack software versions 3.3–4.5, RRID:SCR_014955) of multi-unit spike activity from individual networks were performed under culture conditions (37°C, 5% CO$_2$) and lasted at least 1 hr. Action potentials were detected with a threshold set to −5 standard deviations of the high-pass filtered baseline signal (Butterworth 2$^{nd}$ order high pass filter, 200 Hz cut-off; detection dead time 2 ms).

Raw data from MEA recordings was imported into Matlab using MEA-Tools (*Egert et al., 2002*) and the FIND toolbox (*Meier et al., 2008*). Spontaneous SBEs were detected as follows: Series of spikes with consecutive inter-spike intervals smaller than a threshold value (100 ms) were detected as bursts. SBEs were defined from periods in which a predefined fraction of electrodes showed simultaneous bursts (10% of all sites detecting spikes but minimally 3 and maximally 20 sites to keep criteria comparable between small and large MEAs). To account for buildup and fading phases of SBEs, spikes within a time windows of 25 ms prior to and following this SBE core were included into the SBE. Network activity was characterized by the following parameters: *SBE rate* in the recording period, *SBE strength* as the average number of APs per SBE divided by the number of electrodes with spikes at any time during the recording session (active sites); *AFR* as the grand average firing rate per active site during the recording session. *PFR* was calculated per SBE as the peak of the network-wide firing rate profile (box car filter applied to the global spike train; 0.2 s kernel width)

divided by the number of active sites. *Network synchrony* was determined as average spike train correlation (30 ms bin width) between pairs of active sites.

For the developmental analysis of network activity, recordings from many networks were pooled within time windows of increasing width to account for the slowing development of activity dynamics as networks matured (*Table 2*). Numerical results are presented as mean ± SEM and significance was assessed with a two-tailed independent Student's t-test.

For acute experiments with PTX, we defined as control period the last 1 hr section before application of PTX and excluded the first 10 min after application from the analysis to avoid transients due to handling. To determine the time course of the maturation of inhibition, changes in SBE strength following PTX application were quantified relative to the control period for different DIV. For visualization, trend lines were calculated with a sliding average (±7 DIV).

### Patch-clamp recording and analysis

Patch pipettes (6.3 ± 1.4 MΩ) were filled with a intracellular solution, containing potassium D-gluconate (125 mM; Sigma-Aldrich), KCl (20 mM; Sigma-Aldrich), EGTA (5 mM; Carl Roth), $Na_2$-ATP (2 mM; Carl Roth), HEPES (10 mM; Carl Roth), $MgCl_2$ (2 mM; Sigma-Aldrich) and $CaCl_2$ (0.5 mM; Sigma-Aldrich), adjusted with KOH to pH 7.4, and with sucrose to 320 mOsm. Patch-clamp recordings in whole-cell configuration were conducted at 37°C (PH01 perfusion heating, MCS; TC02 temperature controller, MCS) and perfusion with carbogenated (95% $O_2$ and 5% $CO_2$; Air Liquide, Düsseldorf, Germany) culture medium without horse serum and without Gö6976 and PMA. Data were sampled at 25 kHz (Micro1401 amplifier and Spike2 software; Cambridge Electronics Design Ltd., Cambridge, UK (CED), RRID:SCR_000903). Up to four neurons were recorded sequentially per network for about 30 min each.

Data sets of at least 20 min were analyzed with Matlab. Membrane potential distributions for neurons with resting potentials between −64 ± 4 mV were determined for the entire recording period and averaged across neurons of the same PKC condition.

### Calcium measurements and analyses

To assess neuronal $Ca^{2+}$ dynamics, cultures were transfected with AAV (=Adeno Associated Virus) vectors coding for GCaMP6s (AAV9.CAG.GCaMP6s.WPRE.SV40, titer:~$10^{11}$; Penn Vector Core, School of Medicine Gene Therapy Program, University of Pennsylvania) under control of the CAG promotor after 10–14 days in vitro. $Ca^{2+}$ dynamics were imaged at 20x magnification and 25 Hz frame rate (Examiner Z1 microscope, Zen software 2015, Carl Zeiss, Jena, Germany). Somatic regions were delineated by threshold detection in maximum projections of the $Ca^{2+}$-movie with ImageJ (*Schneider et al., 2012*). The resulting regions of interest were corrected manually. Changes in the $Ca^{2+}$ signal ΔF/F were calculated as relative change to baseline following (*Jia et al., 2011*). For each SBE, the peak of the $Ca^{2+}$ signal (ΔF/F) within 200 ms after onset was related to the PFR determined from simultaneous MEA recordings. The exponential scaling between ΔF/F and PFR was assessed by fitting with the function $\Delta F/F = e^{k*PFR} - 1$ using the Matlab function fminsearch. $Ca^{2+}$ data were derived from five PKC$^N$ and four PKC$^-$ networks at 19–20 DIV in recordings of ~30 min and analyzed with Matlab. $Ca^{2+}$ influx during SBEs was estimated as $e^{0.11*PFR} - 1$ to match the scaling found experimentally. Long-term $Ca^{2+}$ influx was approximated as the $Ca^{2+}$ influx integrated over all SBEs per hour. All results are presented as mean ± SEM. Significance was tested with a two-tailed independent Student's t-test.

## Acknowledgements

We would like to gratefully acknowledge Ute Riede for technical assistance. Special thanks to Philippe Fischer for $Ca^{2+}$ measurements and to Hanna Kuhn and Annika Müller for the patch clamp measurements. Thanks also to Szabina Tudja for some of the MEA recordings and Ad Aertsen for helpful comments on earlier versions of the manuscript. GCaMP6s vectors were kindly provided by the Penn Vector Core, School of medicine Gene Therapy Program at the University of Pennsylvania. Supported by BrainLinks-BrainTools, Cluster of Excellence funded by the German Research Foundation (DFG, grant number EXC 1086) and the German BMBF through the Bernstein Focus Neurotechnology Freiburg*Tuebingen (FKZ 01GQ0420). The article processing charge was funded by the

German Research Foundation (DFG) and the University of Freiburg in the funding programme Open Access Publishing.

## Additional information

### Funding

| Funder | Grant reference number | Author |
|---|---|---|
| Deutsche Forschungsge-meinschaft | EXC 1086 | Ulrich Egert |
| Bundesministerium für Bildung und Forschung | FKZ 01GQ0420 | Ulrich Egert |

The funders had no role in study design, data collection and interpretation, or the decision to submit the work for publication.

### Author contributions
Samora Okujeni, Conceptualization, Data curation, Formal analysis, Validation, Investigation, Visualization, Methodology, Writing—original draft, Writing—review and editing; Ulrich Egert, Conceptualization, Resources, Supervision, Funding acquisition, Methodology, Writing—original draft, Project administration, Writing—review and editing

### Author ORCIDs
Samora Okujeni (iD) https://orcid.org/0000-0001-7924-3651
Ulrich Egert (iD) https://orcid.org/0000-0002-4583-0425

### Ethics
Animal experimentation: Animal handling and tissue preparation were done in accordance with the guidelines for animal research at the University of Freiburg and approved by the Regierungspräsidium Freiburg (permits X-12/08D, X-16/07A, X-15/01H, X-18/04K).

### Decision letter and Author response
Decision letter https://doi.org/10.7554/eLife.47996.030
Author response https://doi.org/10.7554/eLife.47996.031

## Additional files

### Supplementary files
• Transparent reporting form
DOI: https://doi.org/10.7554/eLife.47996.025

### Data availability
Matlab code and source data files are provided for Figures 3-6. Data preprocessing is described in the methods. As the unprocessed data is considerably heavy (over 1TB), the raw data and analysis tools will be provided upon request. Code for the model simulation is available at https://dx.doi.org/10.5281/zenodo.3459678.

The following dataset was generated:

| Author(s) | Year | Dataset title | Dataset URL | Database and Identifier |
|---|---|---|---|---|
| Samora Okujeni, Ulrich Egert | 2019 | Code for Okujeni and Egert, eLife (2019) DOI: 10.7554/eLife.47996 | https://dx.doi.org/10.5281/zenodo.3459678 | Zenodo, 10.5281/zenodo.3459678 |

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
