## [Decision Letter]

Thank you for submitting your article "Self-organization of modular network architecture by activity-dependent neuronal migration and outgrowth" for consideration by *eLife*. Your article has been reviewed by Eve Marder as the Senior Editor, a Reviewing Editor, and three reviewers. The following individuals involved in review of your submission have agreed to reveal their identity: Jordi Soriano (Reviewer #3).

The reviewers have discussed the reviews with one another and the Reviewing Editor has drafted this decision to help you prepare a revised submission.

Summary:

This is a combined experimental and modeling study on neurite growth and migration of cultured neurons. The authors investigate experimentally neural network development with pharmacologically decreased (PKC^-^), normal (PKC^N^) and increased (PKC^+^) neural migration. They find that a higher tendency to migrate leads to higher network clustering but to smaller dendrite sizes and numbers of synaptic partners of a neuron. The results and modelling provide insights into activity-dependent regulation of neuronal growth and are consistent with the notion of an internal set-point in activity.

Essential revisions:

Elements of this work appear to have been reported previously – J Neurosci (2017) and in Front Neurosci (2019). The authors need to fully discuss the relationship of the current findings and study to this previous work, noting any key advances. Abstract concepts such as modularity and clustering need to be unambiguously defined and distinguished. More in-depth analysis of clustering is suggested. There were general concerns about clarity, citations and presentation, outlined in the reviews. These must be addressed.

*Reviewer #1:*

In this manuscript the authors study the development of a network of cortical neurons grown in vitro on multi-electrode arrays (MEAs) allowing to assess many electrophysiological and morphological parameters of the cultures. Cultures were grown in control medium and in the presence of an activator or an inhibitor of Protein Kinase C (PKC). The authors suggest that the activation and inhibition of PKC is stimulating and inhibiting neuronal migration. They find that with PKC stimulation a network with a high degree of neuronal clustering develops. In contrast, with inhibition of PKC, very little neuronal clustering was observed. These findings correspond to simulations based on the van Ooyen model which yield comparable results for network formation with strong and weak neuronal migration. The highly clustered networks show strong spontaneous bursting activity which is rather rare in weakly clustered networks. In contrast, the peak firing rates during the bursts were much higher in weakly clustered networks, suggesting that the net calcium influx was similar in both types of cultures.

The experiments are well done and well documented, and the findings regarding network development with stronger and weaker neuronal migration are very interesting.

There are two issues with this manuscript which, however, could be resolved by the authors with reasonable effort.

1) The manuscript is an extension of previous work by the authors which was published in J Neuroscience previously (Okujeni et al., 2017). In this previous manuscript many of the principle findings reported in the present manuscript were already reported, for example the increased clustering with activation of PKC, the increased spontaneous burst activity with activation of PKC, the increased synaptic density with inhibition of PKC and the different average and peak firing rates with activation and inhibition of PKC. The major novel aspects of the present manuscript are that the development of the network was assessed at different time points revealing the development of the different states. My problem is that the authors do very little to clearly spell out which are new findings and which are extensions or replications of the published work. The authors should reorganize the manuscript such that it becomes clear which data are novel and should much more clearly focus on the novel aspects contained in this manuscript.

2) Based on the simulation experiments, the authors attribute the effects seen with PKC stimulation mostly to the inhibition and promotion of neuronal migration, and interpret the other aspects (as dendritic growth and synaptic density) as a consequence of the change of neuronal migration based on their modified van Ooyen model. While it is attractive to postulate that neuronal migration is the major difference between the three culture conditions, this may not reflect the actual situation. Protein kinase C activity was shown previously to directly affect dendritic development and is known to regulate postsynaptic changes responsible for LTP and LTD in addition to actions on many other target molecules. Reducing the multifaceted effects of PKC to neuronal migration appears to be unrealistic.

*Reviewer #2:*

The paper presents an interesting combined experimental and modeling study on neurite growth and migration of cultured neurons. The authors investigate experimentally neural network development with pharmacologically decreased (PKC^-^), normal (PKC^N^) and increased (PKC^+^) neural migration. They find that a higher tendency to migrate leads to higher network clustering but to smaller dendrite sizes and numbers of synaptic partners of a neuron. Particularly interesting is also the quadratic increase of the number of synapses per neuron with dendrite size, suggesting that axons follow a similar growth rule as dendrites. The authors further analyze the activity of the neuronal cultures and the resulting calcium influx. They find qualitatively different activity for different migration strengths and an exponential dependence of calcium influx on peak firing rate. Remarkably, the long-term calcium influx becomes constant suggesting calcium-dependent homeostasis. On the basis of the experiments, a model is developed that combines an established neuronal outgrowth model with a new model for neuronal migration. It allows simulation of scenarios with decreased, normal and increased neural migration and finds a remarkable similarity in the final network clustering.

While my overall impression of the paper is positive, there are some major questions and points that need to be addressed:

1) An extension of the neuritic field model incorporating neural migration already exists: Eglen, van Oojen and Willshaw (2000). It assumes repulsion between neurons, but is otherwise similar to the present one. It thus needs to be cited and discussed.

2) In the Results section, first the model is presented, then the experiments. According to the Discussion (where the order is reversed), the model has been developed on the basis of the experiments. A reverse order of presentation in the Results section would not be clearer?

3) In the experiments, the calcium influx depends exponentially on the peak firing rate (Figure 6B,D,E). In the model, however, the average firing rate is identified with calcium influx (Figure 2F). What would a plot like Figure 6B,D,E look like for the model dynamics? Might it be suitable to introduce a calcium variable as in Rohrkempter and Abbott (2007) but with a strongly nonlinear accumulation effect? It seems important to carefully relate the migration dynamics in the model (subsection “Network growth model”) to the calcium dynamics suggested by the experiment (Figure 6B,D,E).

4) In the model "connectivity, input activity and firing rates eventually converged to the same levels for different migration conditions" (subsection “Migration and neurite outgrowth shape network architecture”). This point is discussed surprisingly little. Is it not a most severe discrepancy to the experiment? Might it be possible to modify the model with an additional calcium variable to improve it (see point 3)?

5) The dendrite size in the experiments (Figure 3C) shows only a small overshoot, if any. How does this fit with the model dynamics, where the overshoot in neurite size is prominent (Figure 2D)? How does Figure 3J,K fit with the model?

6) It would be good to discuss some of current findings in more detail in the context of previous work on cultured neurons. For example: Figure 5C suggests that there is only an overshoot in activity for PKC^-^ networks. How does this fit with Tetzlaff et al. (2010)? For the axons, previous work has often assumed a growth mechanism opposing that for the dendrites. Does the observation in Figure 3G provide evidence against such a mechanism? Do the authors find critical avalanche dynamics?

7) It is important to discuss how the in vitro results might transfer to in vivo conditions.

*Reviewer #3:*

This is a very interesting study that combines numerical simulations and experiments in in vitro neuronal networks. The study addresses the complex interplay between neuronal spatial arrangement and dynamics due to homeostatic regulation. The manuscript is well structured and written, and it will be definitively of interest for all the scientific community working in neuronal networks, both numerically and experimentally.

Below I list some concerns that should be addressed before publication:

1) In the Introduction, the authors write "we found that developmental clustering boosted SBE". A relatively recent study (Tibau et al., 2018) investigated the impact of aggregation on activity and functional connectivity in neuronal cultures, and pinpointed the importance of neuronal spatial arrangement. Supported by simulations, the authors showed that aggregation promoted activity and connectivity. Since Tibau's study supports the observations of the present work, I think the authors should take a look at it and mention in the Introduction or Discussion section.

2) Although the study of the authors focuses in modularity promoted by self-organization, I think they could mention the efforts in neuroengineering to shape modularity and, in turn, dictate the occurrence of SBE or the richness of activity patterns (see e.g. Bisio et al., 2014; Yamamoto et al., 2018). The fact that aggregation fosters modularity and, in turn, breaks SBE, shaping some sort of spatially-fragmented dynamics, has been observed in engineered networks (Yamamoto et al., 2018) and in self-organized aggregated networks (Teller et al., 2014). These studies can also help enriching the Discussion section.

3) Modularity is a central aspect of the present study. However, the authors do not show any modularity analysis. In the simulations, they can extract the adjacency matrix and then use the freely available Brain Connectivity Toolbox to compute the modularity index Q or similar magnitudes (see the Clustering and Community Structure section of the Brain Connectivity Toolbox). Experimentally, the authors can use the cross-correlation values of neuronal pairs interactions to build a proxy of the functional network and then compute modularity. I am aware that such an analysis may take time. Thus, if they cannot carry it out for the present article, it could be an interesting future direction.

4) In subsection “Simulating activity-dependent neurite growth and migration” (and the Materials and methods section), the authors' model is constructed by seeding neurons on the surface of a torus. I think that it is not clear for the readers the use of such a surface, particularly when trying to compare with the experiments on a 2D flat substrate. I imagine that the torus is used as a 'mathematical construction' to promote local connectivity, but it needs clarification and more details.

5) Related to this, the authors should be aware of the work by Hernández-Navarro et al. (2017), who presented a model and numerical simulations on the impact of neuronal aggregation and neurite length in shaping network connectivity, and the importance of spatial correlations inherited by neuronal spatial proximity. I think the authors should refer to that study, e.g. in the context of the explanations provided in the Results section.

6) In subsection “Migration and neurite outgrowth shape network architecture”, the authors introduce the Clustering Index CI for the first time. They need to define CI here and explain what it means. The fact that CI=1 corresponds to a homogeneous system (and 0 to a clustered) is very confusing, since "clustering index" itself suggests the opposite. The authors should at least clarify the meaning of CI and its range of variability as soon as possible in the text. I also note that the authors also use the term "degree of clustering" (e.g. in subsection “Differences in PFR reflect variations of network recruitment during SBEs”) which is 1-CI. Overall, reading can be difficult.

7) In subsection "Clustering promoted SBE initiation and increased AFRs" the authors explain the link between SBE and aggregation. This relation is very interesting. I wonder whether the experimental data can provide evidences of important differences in the spatial foci of initiation (e.g. as in Orlandi et al., 2013) in homogeneous/aggregated networks, or in the structure of propagating fronts. Can the authors elaborate on this interesting link between activity initiation and aggregation? How the structure of the spatiotemporal activity fronts change with aggregation?

8) Related to this, the authors state that SBE decreases with aggregation. This is interesting. Have the authors observed a transition from whole-network activation to partial dynamics (e.g. as in Yamamoto et al., 2018 or Teller et al., 2014)?

---

## [Author Response]

Reviewer #1:[…]1) The manuscript is an extension of previous work by the authors which was published in J Neuroscience previously (Okujeni et al., 2017). In this previous manuscript many of the principle findings reported in the present manuscript were already reported, for example the increased clustering with activation of PKC, the increased spontaneous burst activity with activation of PKC, the increased synaptic density with inhibition of PKC and the different average and peak firing rates with activation and inhibition of PKC. The major novel aspects of the present manuscript are that the development of the network was assessed at different time points revealing the development of the different states. My problem is that the authors do very little to clearly spell out which are new findings and which are extensions or replications of the published work. The authors should reorganize the manuscript such that it becomes clear which data are novel and should much more clearly focus on the novel aspects contained in this manuscript.

In the current study, we focused on the homeostatic development of the different network architectures described in Okujeni et al. (2017) and Okujeni et al. (2019). Building on the well-known model by van Ooyen et al. and preliminary observations in our cell cultures, we assessed the morphology and dynamics at several time points in the course of development and identified the interaction between neuronal migration and neurite outgrowth as a key mechanism shaping networks with connectivity ranging from clustered to homogeneous. Moreover, we propose that due to a supra-linear calcium influx depending on neuronal depolarization and the network PFR, long-term calcium influx could converge in the different networks despite of their differences in AFR and burst rates. Here, we change the perspective of previous works on homeostatic network growth that assumed neuronal spikes as a sole source of calcium and hence proposed the spike rate as the control parameter.

We revised the manuscript to clarify which aspects are novel and which ones have been published already.

In the Introduction we now point out which findings were already presented in our previous papers and highlight novel aspects of the current study.

Previous results are also clarified in Results sectionsubsection “Mesoscale architecture and the development of spontaneous activity” and subsection “Growth and migration shape the framework for synaptic connectivity” and Figure 4 was moved into the supplement (now Figure 4—figure supplement 1) since it was intended only to show readers not familiar with the older papers what the activity we find typically looks like.

We made major changes to the Discussion section now focusing more on the new findings.

2) Based on the simulation experiments, the authors attribute the effects seen with PKC stimulation mostly to the inhibition and promotion of neuronal migration, and interpret the other aspects (as dendritic growth and synaptic density) as a consequence of the change of neuronal migration based on their modified van Ooyen model. While it is attractive to postulate that neuronal migration is the major difference between the three culture conditions, this may not reflect the actual situation. Protein kinase C activity was shown previously to directly affect dendritic development and is known to regulate postsynaptic changes responsible for LTP and LTD in addition to actions on many other target molecules. Reducing the multifaceted effects of PKC to neuronal migration appears to be unrealistic.

We certainly agree with the reviewer that PKC influences a range of intracellular processes, with its influence on the cytoskeleton being only one of them. The model incorporates this only indirectly since we do not model the molecular processes in any detail. On the one hand, the direct action of PKC on dendritic development and mobility is simplified as we address the relative contribution of outgrowth and migration by modifying the migration rate only. Because of the feedback loop, however, this also affects the amount of outgrowth. On the other hand, outgrowth in fact creates only *potential* connections, but does not define their strength.

Functionally, the model takes the perspective of effective connectivity – which clearly could incorporate synaptic plasticity. In fact, differences in synaptic weights (or impact, to take into account multapses) showed as differences in the EPSP size distributions for different networks reported in Okujeni et al. (2017). We therefore cannot take the physical dimensions of the ‘connectivity fields’ in the model at face value but rather the relative contributions of growth and migration to connectivity regulation. We now explain this in more detail in the Discussion section.

Reviewer #2:[…]1) An extension of the neuritic field model incorporating neural migration already exists: Eglen, van Oojen and Willshaw (2000). It assumes repulsion between neurons, but is otherwise similar to the present one. It thus needs to be cited and discussed.

Thank you very much for this comment, unfortunately we overlooked this paper. We now address the differences to our model in the Discussion section.

2) In the Results section, first the model is presented, then the experiments. According to the Discussion section (where the order is reversed), the model has been developed on the basis of the experiments. A reverse order of presentation in the Results section would not be clearer?

In the Results section, we first present the model to introduce and motivate the overall concept of interacting neurite growth and migration and clustering. We believe that this makes it easier for the reader to embed the different in vitro results into a coherent picture. It actually also reflects the development of our own ideas on how these networks self-organize.

We revised the discussion and now discuss the model alongside with our in vitro results.

3) In the experiments, the calcium influx depends exponentially on the peak firing rate (Figure 6B,D,E). In the model, however, the average firing rate is identified with calcium influx (Figure 2F). What would a plot like Figure 6B,D,E look like for the model dynamics? Might it be suitable to introduce a calcium variable as in Rohrkempter and Abbott (2007) but with a strongly nonlinear accumulation effect? It seems important to carefully relate the migration dynamics in the model (subsection “Network growth model”) to the calcium dynamics suggested by the experiment (Figure 6B,D,E).

In the model, firing rate (and thus [Ca^2+^]_i_) is derived based on a sigmoidal gain function and the input. We do not model calcium dynamics explicitly, e.g., by introducing a non-linear mapping of firing rate on Ca^2+^. Such a modification would not change the general growth dynamics but just rescale the gain function. To include the observed dependence of Ca^2+^ on PFR it would be necessary to introduce a spiking model (as e.g. Rohrkempter and Abbott). However, simulating network development across weeks at the millisecond resolution required to model the network dynamics is still impractical. In the Rohrkempter and Abbott, the growth rate was increased considerably during early development and reduced to more realistic rates only when the network was close to equilibrium. This approach is, however, not applicable for us as migration and neurite outgrowth interact gradually.

We now address these issues in the Discussion section.

4) In the model "connectivity, input activity and firing rates eventually converged to the same levels for different migration conditions" (subsection “Migration and neurite outgrowth shape network architecture”). This point is discussed surprisingly little. Is it not a most severe discrepancy to the experiment? Might it be possible to modify the model with an additional calcium variable to improve it (see point 3)?

The model converges because it assumes stationary neuronal firing rates for a given connectivity state. We consider it as one of the main findings that firing rate is not a good proxy for the homeostatic regulation with highly fluctuating spike rates – as in bursts. The model, as in earlier publications by others, already assumes that network activity actually converges towards a certain Ca^2+^ set-point and in this respect, there is no discrepancy to the in vitro results. Bursting dynamics and the dependence of Ca^2+^ influx on PFRs (and the resulting depolarization) are not modelled and would require a spiking neuron model (see answer to question 3).

We now address these aspects in the Discussion section.

5) The dendrite size in the experiments (Figure 3C) shows only a small overshoot, if any. How does this fit with the model dynamics, where the overshoot in neurite size is prominent (Figure 2D)? How does Figure 3J,K fit with the model?

In the model, the overshoot depends on the membrane time constant for activity integration and the steepness of the sigmoidal transfer function mapping input to firing rate. Only a steep sigmoid produces a sudden onset of “network activity” once critical connectivity levels are attained during development. Shallower functions lead to saturation rather than to overshoot.

We show this dependence in Figure 2—figure supplement 2 and address the issue in the Discussion section. Maximal connectivity (based on presynaptic bouton counts) cannot account for potential multapses, which likely depend on CI, and synaptic plasticity/homeostatic synaptic scaling in mature networks (Okujeni et al., 2017). We interpret the combination of synaptic efficacy and number of synapses as effective connectivity, which is what the model represents. Maximal connectivity was based on the number of synapses and thus reflects structural connectivity only.

We now address such a mechanism and its implications for the concept of homeostatic growth in the Discussion section.

6) It would be good to discuss some of current findings in more detail in the context of previous work on cultured neurons. For example: Figure 5C suggests that there is only an overshoot in activity for PKC^-^ networks. How does this fit with Tetzlaff et al. (2010)? For the axons, previous work has often assumed a growth mechanism opposing that for the dendrites. Does the observation in Figure 3G provide evidence against such a mechanism? Do the authors find critical avalanche dynamics?

In the Introduction, we now mention studies supporting our previous results.

We further compare our connectivity estimates with previous studies (subsection “Growth and migration shape the framework for synaptic connectivity”) and refer to pervious reports on cell culture development regarding the growth overshoot (subsection “Migration contributes to homeostatic network development”).

Previous studies assumed an antagonistic regulation of axonal and dendritic growth. Our results suggest, however, that both neurite types are regulated in the same direction. We now refer to our previous report (Okujeni et al., 2017) and to the modeling paper by Tetzlaff et al. (2010) (subsection “Growth and migration shape the framework for synaptic connectivity”).

In Tetzlaff et al. (2010), the avalanche analysis at the time indeed yielded no significant differences between PKC^-^ and PKC^N^ networks and the data was pooled. We now have a much larger database and some findings may make a difference with respect to those in Tetzlaff et al. While it would certainly be relevant to redo that analysis with the new data set, more recent theoretical studies have shown additionally that undersampling and other aspects of such analyses can be problematic in assessing criticality (Levina et al., 2017; Wilting et al., 2018). Addressing these issues would require changes to the methods used in Tetzlaff et al. and a quite extensive discussion. We think therefore that criticality analyses would be beyond the scope of the current paper.

7) It is important to discuss how the in vitro results might transfer to in vivo conditions.

We now refer to the in vivo situation with implications of altered migration for pathological brain states in the conclusion section. We prefer, however, not to expand too much on this to avoid overinterpretation of our results.

Reviewer #3:[…]1) In the Introduction, the authors write "we found that developmental clustering boosted SBE". A relatively recent study (Tibau et al., 2018) investigated the impact of aggregation on activity and functional connectivity in neuronal cultures, and pinpointed the importance of neuronal spatial arrangement. Supported by simulations, the authors showed that aggregation promoted activity and connectivity. Since Tibau's study supports the observations of the present work, I think the authors should take a look at it and mention in the Introduction or Discussion section.

This is indeed an important study in the context of this manuscript and its preceding papers, since it supports many of our results – with an approach to manipulate network architecture without pharmacological intervention. Thank you very much for the remark. We now mention the paper in the Introduction.

2) Although the study of the authors focuses in modularity promoted by self-organization, I think they could mention the efforts in neuroengineering to shape modularity and, in turn, dictate the occurrence of SBE or the richness of activity patterns (see e.g. Bisio et al., 2014; Yamamoto et al., 2018). The fact that aggregation fosters modularity and, in turn, breaks SBE, shaping some sort of spatially-fragmented dynamics, has been observed in engineered networks (Yamamoto et al., 2018) and in self-organized aggregated networks (Teller et al., 2014). These studies can also help enriching the Discussion section.

Thank you very much for reminding us of these papers. They nicely support our previous results shown in Okujeni et al. (2017), where we had a strong focus on structure-function dependencies in clustered and homogeneous networks. In the current study, however, our main focus is laid on the growth mechanisms that lead to different network architectures. Nevertheless, since these studies nicely support many aspects of our findings, we now mention Bisio, Teller and Yamamoto in the Introduction.

3) Modularity is a central aspect of the present study. However, the authors do not show any modularity analysis. In the simulations, they can extract the adjacency matrix and then use the freely available Brain Connectivity Toolbox to compute the modularity index Q or similar magnitudes (see the Clustering and Community Structure section of the Brain Connectivity Toolbox). Experimentally, the authors can use the cross-correlation values of neuronal pairs interactions to build a proxy of the functional network and then compute modularity. I am aware that such an analysis may take time. Thus, if they cannot carry it out for the present article, it could be an interesting future direction.

Thank you for the link to the toolbox. We now applied the measure to the connectivity matrices of the simulation, which shows that modularity increases with migration (and clustering).

Yet, although modularity analyses of the in vitro data shows the same trend (see Author response image 1), we think that a reconstruction of connectivity (correlation matrices) from MEA recordings to assess modularity is not robust. This is because the intra- vs. inter-cluster connectivity is not accessible with the MEA as the electrode pitch is close to the inter-cluster distance (in contrast to Ca-imaging) and in clustered networks we only have one electrode per cluster. In consequence, we would only see the connectivity between the sampled “modules” in PKC^+^ networks and implicitly the definition of modularity would change. We therefore decided not to emphasize this aspect quantitatively in the current manuscript.

Preliminary analysis of (Louvain) modularity in mature networks based on MEA data and pairwise correlation matrices of spike trains at active sites (autocorrelation was set to zero; bin width 10 ms, at least 100 electrodes with spike activity, N = 31, 40, 12 networks, p = 1.1*10^-2^ for PKC^-^ vs. PKC^N^, p = 1.0*10^-6^ for PKC^+^ vs. PKC^N^).

4) In subsection “Simulating activity-dependent neurite growth and migration” (and Materials and methods section), the authors' model is constructed by seeding neurons on the surface of a torus. I think that it is not clear for the readers the use of such a surface, particularly when trying to compare with the experiments on a 2D flat substrate. I imagine that the torus is used as a 'mathematical construction' to promote local connectivity, but it needs clarification and more details.

A torus was used to avoid boundary conditions and to simulate the situation of neurons in the center of the much larger in vitro networks of 200.000 neurons. We state this now in subsection “Network growth model”.

5) Related to this, the authors should be aware of the work by Hernández-Navarro et al. (2017), who presented a model and numerical simulations on the impact of neuronal aggregation and neurite length in shaping network connectivity, and the importance of spatial correlations inherited by neuronal spatial proximity. I think the authors should refer to that study, e.g. in the context of the explanations provided in the Results section.

We now mention this paper in the Introduction.

6) In subsection “Migration and neurite outgrowth shape network architecture”, the authors introduce the Clustering Index CI for the first time. They need to define CI here and explain what it means. The fact that CI=1 corresponds to a homogeneous system (and 0 to a clustered) is very confusing, since "clustering index" itself suggests the opposite. The authors should at least clarify the meaning of CI and its range of variability as soon as possible in the text. I also note that the authors also use the term "degree of clustering" (e.g. in subsection “Differences in PFR reflect variations of network recruitment during SBEs”) which is 1-CI. Overall, reading can be difficult.

We added a short description of CI in subsection “Migration and neurite outgrowth shape network architecture” and subsection “Morphological analyses”.

7) In subsection "Clustering promoted SBE initiation and increased AFRs" the authors explain the link between SBE and aggregation. This relation is very interesting. I wonder whether the experimental data can provide evidences of important differences in the spatial foci of initiation (e.g. as in Orlandi et al., 2013) in homogeneous/aggregated networks, or in the structure of propagating fronts. Can the authors elaborate on this interesting link between activity initiation and aggregation? How the structure of the spatiotemporal activity fronts change with aggregation?

We have addressed this issue in a recent paper (Okujeni et al., 2019), where we also refer to Orlandi et al. (2013).

8) Related to this, the authors state that SBE decreases with aggregation. This is interesting. Have the authors observed a transition from whole-network activation to partial dynamics (e.g. as in Yamamoto et al., 2018 or Teller et al., 2014)?

Yes, we indeed find nice correspondence to the Yamamoto and Teller studies that we now mention in the current manuscript. In Okujeni et al. (2017), we showed already that weaker burst strength and depolarization of neurons in clustered networks is associated with reduced network recruitment during SBEs (Figure 6A). This aspect partly reflects the network architecture but was also caused by the high rate of SBE generation and associated refractoriness or synaptic depression.